# Benchmarking LLMs via Uncertainty Quantification

**Fanghua Ye**[1,2]  **Mingming Yang**[1]  **Jianhui Pang**[1,3]  **Longyue Wang**[1,*]
**Derek F. Wong**[3]  **Emine Yilmaz**[2]  **Shuming Shi**[1]  **Zhaopeng Tu**[1]
[1]Tencent AI Lab   [2]University College London   [3]University of Macau
fanghua.ye.19@ucl.ac.uk, nlp2ct.pangjh3@gmail.com
derekfw@um.edu.mo, emine.yilmaz@ucl.ac.uk
{shanemmyang, vinnylywang, shumingshi, zptu}@tencent.com

## Abstract

The proliferation of open-source Large Language Models (LLMs) from various institutions has highlighted the urgent need for comprehensive evaluation methods. However, current evaluation platforms, such as the widely recognized HuggingFace open LLM leaderboard, neglect a crucial aspect – **uncertainty**, which is vital for thoroughly assessing LLMs. To bridge this gap, we introduce a new benchmarking approach for LLMs that integrates uncertainty quantification. Our examination involves nine LLMs (LLM series) spanning five representative natural language processing tasks. Our findings reveal that: I) *LLMs with higher accuracy may exhibit lower certainty*; II) *Larger-scale LLMs may display greater uncertainty compared to their smaller counterparts*; and III) *Instruction-finetuning tends to increase the uncertainty of LLMs*. These results underscore the significance of incorporating uncertainty into the evaluation of LLMs. Our implementation is available at `https://github.com/smartyfh/LLM-Uncertainty-Bench`.

## 1   Introduction

Large Language Models (LLMs) have gained significant traction within both academia and industry, with numerous organizations and companies open-sourcing their versions of LLMs [9, 74, 36, 68]. LLMs are highly versatile, demonstrating capabilities in various tasks such as question answering, document summarization, dialogue systems, and machine translation [70, 52]. Given the growing interest and advancements in LLMs, it is crucial to establish appropriate methods for evaluating their performance [42, 71, 9]. However, conducting a comprehensive evaluation of LLMs remains a challenging endeavor [28, 75].

To address this challenge, several open leaderboards such as the popular HuggingFace open LLM leaderboard,[2] OpenCompass [14], Chatbot Arena [75], and FlagEval [6] have emerged, providing a comparative analysis of LLM performance. Despite their usefulness, these leaderboards possess a significant limitation: *They do not take into account the uncertainty of LLMs*. For example, the HuggingFace open LLM leaderboard only utilizes accuracy as the evaluation metric. However, as demonstrated in Figure 1, two LLMs may achieve identical accuracy scores but exhibit different levels of uncertainty regarding the question. This is analogous to students taking exams of multiple-choice questions, where two students may select the same answer but actually possess distinct degrees of uncertainty or comprehension about the question. Consequently, it is necessary to incorporate uncertainty into the evaluation process to achieve a more comprehensive assessment of LLMs.

In this paper, we propose the utilization of conformal prediction [64, 5] as the method to quantify uncertainty in LLMs. Compared to alternative methods such as Bayesian variational inference [30],

---

*Corresponding author.

[2]`https://huggingface.co/spaces/HuggingFaceH4/open_llm_leaderboard`

conformal prediction offers multiple advantages including ease of implementation, high efficiency, distribution-free and model-agnostic, and a statistically **rigorous** estimation of uncertainty rather than a heuristic approximation [5]. Hence, conformal prediction can serve as a practical and principled means for assessing the uncertainty of LLMs.

Specifically, we benchmark nine open-source LLMs (LLM series) across five typical Natural Language Processing (NLP) tasks, namely question answering, reading comprehension, commonsense inference, dialogue response selection, and document summarization. Given that most existing open leaderboards and benchmarking datasets [45] focus on multiple-choice tasks, we also adopt the multiple-choice question setting for all tasks. Although some of these tasks (e.g., document summarization) are inherently generative, it is challenging to develop a deterministic and reproducible method for quantifying the uncertainty within the generated text due to randomness in the generation process. Instead, we convert all tasks into multiple-choice questions, with the objective of each task being to select the correct option from the provided choices. Our empirical results reveal the following observations: I) *LLMs demonstrating higher accuracy may exhibit lower certainty*; II) *LLMs with larger scales may display greater uncertainty than their smaller counterparts*; and III) *LLMs after instruction-finetuning tend to possess higher uncertainty*.

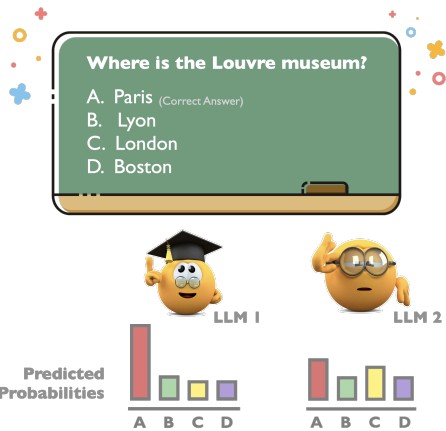

Figure 1: An illustration of two LLMs accurately predicting the true answer (with option A possessing the highest probability), but showing different levels of uncertainty. Note that when both LLMs predict a wrong answer, they may also display different levels of uncertainty.

## 2 Related work

**Uncertainty Quantification** Uncertainty quantification [22, 2, 25] has been an active area of research in both machine learning and NLP due to its importance in real-world applications such as decision making, risk assessment, human-AI collaboration, and so on. Typical uncertainty quantification methods include confidence-based methods [30], Bayesian methods [39], and ensemble methods [54]. Confidence-based methods such as entropy can be sensitive to poor calibration and may not fully capture models' underlying uncertainties [48]. Bayesian methods and ensemble methods usually suffer from high computational complexity [25], making them not suitable for assessing the uncertainty of LLMs.

**Conformal Prediction** Recently, there has been a growing interest in applying conformal prediction for uncertainty quantification [5, 38, 53, 46]. For example, conformal prediction has been applied to part-of-speech prediction [17], paraphrase detection [26], and fact verification [21]. Similar to the process of elimination used by students during exams, conformal prediction identifies a subset of potential labels in classification tasks by excluding improbable labels, which is statistically guaranteed to contain the true label, and quantifies uncertainty as the size of this subset [4]. The coverage guarantee property makes conformal prediction a highly robust uncertainty quantification method. In addition, conformal prediction is non-parametric, distribution-free (i.e. *not dependent on any specific distributional assumptions about the data*), model-agnostic, and computationally efficient [5]. Therefore, it is a favorable choice in the context of LLMs.

**LLM Evaluation** Evaluating the performance of LLMs is a crucial aspect of their development and deployment [32]. Current studies assess LLMs from different angles using specific datasets, such as MMLU [29] for knowledge, HellaSwag [72] for reasoning, HaluEval [40] for hallucination, GSM8K [13] for math, and BOLD [18] for fairness. Besides, evaluation platforms like HuggingFace open LLM leaderboard and Chatbot Arena [75] have also been developed to facilitate comparisons among LLMs. Despite these efforts, the critical aspect of uncertainty in LLMs remains underexplored. More recently, some research has begun to consider uncertainty in LLMs [67, 69, 43, 10, 65, 20]. However, these approaches such as the sampling-based semantic entropy [37] are heuristic and lack

a standardized methodology for benchmarking purposes. In contrast, our utilization of conformal prediction can provide a robust and systematic evaluation of uncertainty.

## 3 Background of conformal prediction

Conformal prediction serves as a **distribution-free** and **model-agnostic** approach to uncertainty quantification [64, 8, 5, 23]. It can transform any heuristic notion of uncertainty from any model into a statistically **rigorous** one. As aforementioned, for multi-class classification tasks, conformal prediction outputs a prediction set of possible labels (answers) that encompasses the correct label with a user-specified error rate and expresses uncertainty as the set size. Intuitively, a larger set size indicates higher uncertainty and vice versa.

Formally, let $f$ be a model that classifies an input $X$ into $K$ pre-defined classes, represented by $\mathcal{Y} = \{1, \ldots, K\}$. To measure the uncertainty of $f$, for any given test instance $X_t$ and its corresponding true label $Y_t$, conformal prediction produces a prediction set of labels $\mathcal{C}(X_t) \subset \mathcal{Y}$ such that

$$p(Y_t \in \mathcal{C}(X_t)) \geq 1 - \alpha, \tag{1}$$

where $\alpha \in (0, 1)$ is a user-specified error rate.

Equation (1) requires that the generated prediction set should contain the true label $Y_t$ with a probability no smaller than $1 - \alpha$. This coverage guarantee requirement can be achieved with the aid of a small amount of held-out *calibration data* $\mathcal{D}_{cal} = \{(X_c^{(1)}, Y_c^{(1)}), \ldots, (X_c^{(n)}, Y_c^{(n)})\}$, where $n$ denotes the number of data points in the calibration set.[3] More specifically, conformal prediction works in the following process [5] to create the prediction set:

1. Identify a heuristic notion of uncertainty based on the model $f$;
2. Define a conformal score function $s(X, Y) \in \mathbb{R}$ with larger scores encoding worse agreement between $X$ and $Y$;
3. Compute conformal scores on the calibration set $s_1 = s(X_c^{(1)}, Y_c^{(1)}), \ldots, s_n = (X_c^{(n)}, Y_c^{(n)})$ and calculate a threshold $\hat{q}$ as the $\frac{\lceil (n+1)(1-\alpha) \rceil}{n}$ quantile of the calibration scores,

$$\hat{q} = \text{quant}\left(\{s_1, \ldots, s_n\}, \frac{\lceil (n+1)(1-\alpha) \rceil}{n}\right), \tag{2}$$

   where $\lceil \cdot \rceil$ is the ceiling function;
4. Construct the prediction set for each test instance $X_t$ as

$$\mathcal{C}(X_t) = \{Y' \in \mathcal{Y} : s(X_t, Y') \leq \hat{q}\}. \tag{3}$$

For classification tasks, it is a common choice to adopt the softmax score (i.e. estimated probability of each class by the model) as the *heuristic* notion of uncertainty. However, this score usually does not reflect the true class distribution due to over-confident or under-confident model predictions. In this work, we consider two conformal score functions to convert the softmax score to a *statistically rigorous* notion of uncertainty (which is calibrated in the sense that the prediction sets satisfy the coverage guarantee requirement).

**Least Ambiguous set-valued Classifiers (LAC)**    LAC [58] defines the conformal score function as

$$s(X, Y) = 1 - f(X)_Y, \tag{4}$$

where $f(X)_Y$ is the softmax score corresponding to the true label. It has been proven that LAC can lead to prediction sets with the smallest average size [58]. However, it may undercover hard instances and overcover easy ones.

**Adaptive Prediction Sets (APS)**    APS [57] defines the conformal score function as

$$s(X, Y) = \sum_{\{Y' \in \mathcal{Y} : f(x)_{Y'} \geq f(x)_Y\}} f(X)_{Y'}, \tag{5}$$

where $f(X)_{Y'}$ represents the softmax score corresponding to the label $Y' \in \mathcal{Y}$. Equation (5) is equivalent to summing the ranked scores of each label, from the higher to the lower, until reaching

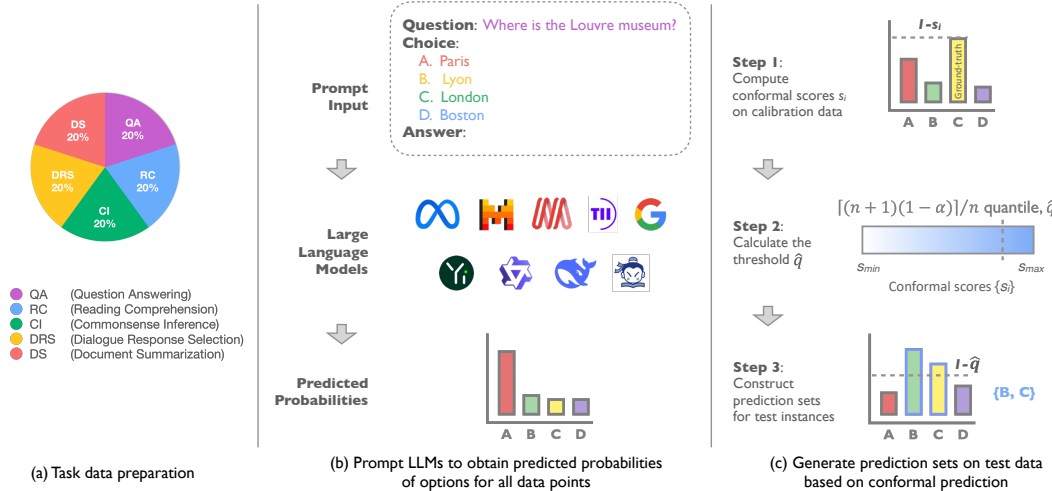

(a) Task data preparation

(b) Prompt LLMs to obtain predicted probabilities of options for all data points

(c) Generate prediction sets on test data based on conformal prediction

Figure 2: The overall process of applying conformal prediction for uncertainty quantification in LLMs. (a) Five distinct tasks are considered, and a dataset comprising 10,000 instances is prepared for each task. (b) Each data instance is transformed into a multiple-choice question, and nine LLMs (LLM series) are prompted to generate predicted probabilities for the given options. (c) Each dataset is divided into a calibration set and a test set, followed by the application of conformal prediction to generate prediction sets for test set instances. For illustrative purposes, demonstrations in the prompt input are excluded, and solely the process of constructing prediction sets utilizing the LAC conformal score function is demonstrated. In addition, only four options of the question are presented.

the true label. Compared to LAC, APS leverages the softmax scores of all labels, not just the true label. It addresses the limitation of LAC but suffers from, on average, larger prediction sets.

The overall process of employing conformal prediction for uncertainty quantification in LLMs is illustrated in Figure 2. In the following sections, we first elucidate on the evaluation tasks and their associated datasets, then provide details about the evaluation prompts used to extract softmax scores (i.e. predicted probabilities) from LLMs, and finally, introduce the adopted evaluation metrics.

## 4 Evaluation tasks and datasets

LLMs have demonstrated remarkable capabilities across various aspects [28, 50]. It is essential to develop multiple tasks to evaluate their performance comprehensively. For this purpose, we consider five typical NLP tasks, including question answering, reading comprehension, commonsense inference, dialogue response selection, and document summarization. For each task, we prepare a dataset with 10,000 instances. In addition, we formulate each task as a Multiple-Choice Question Answering (MCQA) task and the objective is to select the *only* correct answer out of six possible options (i.e. A, B, C, D, E, and F). It is worth emphasizing that the prevailing benchmarking open leaderboards and datasets also focus on MCQA tasks [28, 45].

**Question Answering (QA)**   QA is applied to evaluate an LLM's proficiency in utilizing its extensive world knowledge to provide answers to a diverse range of questions. For this task, we adopt **MMLU** [29] as the evaluation dataset. MMLU encompasses a total of 57 subjects, spanning various disciplines such as elementary mathematics, US history, computer science, and law. These subjects are further classified into four broad categories, namely humanities, social sciences, STEM, and others (business, health, misc.). For each category, we sample 2500 instances, leading to 10,000 instances in total.

**Reading Comprehension (RC)**   RC is used for testing an LLM's ability to understand and analyze a given context, comprehend the meaning of words and sentences, and answer questions based

---

[3]It is also required that the test data points are drawn from the same distribution as the calibration data.

on the information presented in the context. It also tests the ability of LLMs to make inferences and draw conclusions from the given context. We take **CosmosQA** [31] as the evaluation dataset. CosmosQA focuses on *reading between the lines* [51] over a diverse collection of people's everyday narratives that require reasoning beyond the exact text spans in the context. Due to the unavailability of ground truth labels for the test set in CosmosQA, we sample 10,000 instances from the training and development sets.

**Commonsense Inference (CI)** CI is leveraged to evaluate the ability of LLMs to understand and reason about the relationships between concepts and events based on commonsense and background knowledge. This type of testing helps to assess the generalization and reasoning abilities of LLMs beyond simple pattern recognition tasks. We employ **HellaSwag** [72] as the evaluation dataset. HellaSwag focuses on *commonsense natural language inference* whose target is to select the most likely followup to a given event description. Same as CosmosQA, we sample 10,000 instances from the training and development sets of HellaSwag for the purpose of evaluation.

**Dialogue Response Selection (DRS)** DRS is adopted for assessing the ability of LLMs to comprehend the meaning of a given dialogue and select an appropriate response from a set of possible responses. This includes the ability to comprehend the meaning of the user's input, select appropriate responses that are relevant to the conversational context, and maintain coherence and consistency in the dialogue. We utilize the dialogue data from the HaluEval [40] benchmark as the evaluation dataset (denoted as **HaluDial**). This dataset consists of exactly 10,000 instances and is built upon OpenDialKG [49], a knowledge-grounded dialogue dataset.

**Document Summarization (DS)** DS is taken to evaluate the proficiency of LLMs in comprehending the substance and context of a given document, and in producing a succinct and cohesive summary that effectively conveys the crucial information and main ideas of the document. This requires the LLM to have a good understanding of the language, the topic, and the structure of the document. Similar to DRS, we adopt the summarization data from the HaluEval [40] benchmark as the evaluation dataset (denoted as **HaluSum**). This dataset also comprises precisely 10,000 instances and is derived from CNN/Daily Mail [59], a summarization dataset pertaining to news articles.

Out of the aforementioned five datasets, MMLU, CosmosQA, and HellaSwag originally consisted of questions with four options each, while HaluDial and HaluSum had only two options per question. To standardize the number of options, two additional choices were added to each question in HaluDial and HaluSum by randomly selecting from the choices of other questions in HaluDial and HaluSum, respectively. Furthermore, all datasets were modified to include two more options, "*I don't know*" and "*None of the above*", resulting in a total of six possible options for each question.

## 5   Evaluation prompts and metrics

As delineated in § 3, one crucial step of leveraging conformal prediction for uncertainty quantification is to acquire the softmax score corresponding to each option. Next, we explicate the method used to elicit these scores from LLMs, followed by a description of the evaluation metrics utilized.

**Prompting Strategies** Following previous works [29, 24, 73, 12, 75], we rely on prompt engineering rather than supervised finetuning as the testing approach to evaluating the performance of LLMs on each task. However, our preliminary results show that LLMs are sensitive to prompts. In this regard, we consider three prompting strategies, including *Base Prompt*, *Shared Instruction Prompt*, and *Task-specific Instruction Prompt*, to reduce the influence of LLMs' sensitivity to different prompts, thereby ensuring a fairer comparison.

- **Base Prompt:** This strategy directly combines the question and all of its options as the input and prompts the LLM to output the correct option with a prefix "Answer:".
- **Shared Instruction Prompt:** This strategy adds a general description of the task before the question, informing the LLM that the task is to solve a multiple-choice question and there is only one correct answer out of six options.
- **Task-specific Instruction Prompt:** Instead of using a shared instruction, this strategy provides a task-specific instruction that briefly describes the task and the expected type of option.

These prompting strategies facilitate a systematic and standardized evaluation of the performance of LLMs. The prompt templates linked with each strategy are elaborated in Appendix D. The softmax score for each prompt is derived by subjecting the logits corresponding to each option letter (i.e. A, B, C, D, E, and F) to the softmax function. The said logits are generated by the language modeling head in contemporary causal LLMs. It is worth noting that only the logits associated with the last token of the prompt input are utilized.

**Evaluation Metrics** We evaluate LLMs from two perspectives, namely prediction accuracy and prediction uncertainty. For prediction accuracy, we adopt the commonly used metric – Accuracy (**Acc**). To evaluate prediction uncertainty, we use Set Size (**SS**), which is a primary metric for conformal prediction [5]. Let $Y_p$ be the prediction for the test instance $(X_t, Y_t) \in \mathcal{D}_{test}$. These two metrics can be calculated as follows:

$$Acc = \frac{1}{|\mathcal{D}_{test}|} \sum_{(X_t,Y_t)\in\mathcal{D}_{test}} \mathbb{1}(Y_p = Y_t), \quad SS = \frac{1}{|\mathcal{D}_{test}|} \sum_{(X_t,Y_t)\in\mathcal{D}_{test}} |\mathcal{C}(X_t)|, \quad (6)$$

where $\mathbb{1}(\cdot)$ is the indicator function.

In addition to Acc and SS, we report the Coverage Rate (**CR**) to verify if the coverage guarantee requirement shown in Eq. (1) has been satisfied. The CR metric is calculated as

$$CR = \frac{1}{|\mathcal{D}_{test}|} \sum_{(X_t,Y_t)\in\mathcal{D}_{test}} \mathbb{1}(Y_t \in \mathcal{C}(X_t)). \quad (7)$$

# 6 Evaluation results

## 6.1 Setup

In our experiments, we set the error rate $\alpha$ to 0.1, implying that the prediction set should include the true label with a probability of at least 0.9. In order to better excite the ability of LLMs, we incorporate examples or demonstrations in the prompt, adhering to the in-context learning paradigm [19]. Specifically, we provide five demonstrations for QA, RC, and CI tasks. For the DRS task, we utilize three demonstrations, while we use only one demonstration for the DS task due to constraints on input length and inference cost. The maximum input length for all tasks is set to 2048 tokens. In addition, for each task, we allocate 50% of the data as the calibration set and the remaining 50% as the test set. We report results on the test set. These results represent the average value obtained from the two conformal score functions, namely, LAC and APS, as well as the three prompting strategies. It is noteworthy that while both LAC and APS fulfill the coverage guarantee requirement, they may produce prediction sets of varying sizes. By taking the average value of these two score functions, we aim to mitigate the influence of different score functions on the evaluation of uncertainty, thereby ensuring a more rigorous and reliable assessment.

## 6.2 Evaluated models

We select a diverse set of nine representative models (or model series) from the vast array of open-source LLMs available. These models encompass various architectures and training methodologies, thereby allowing for a comprehensive benchmarking analysis. Specifically, the chosen models include the Llama-2 series [63], Mistral-7B [34], Falcon series[4] [3], MPT-7B [62], Gemma-7B [60], Qwen series [7], Yi series [1], DeepSeek series [15], and InternLM-7B [61]. For all models, we utilize their checkpoints from the HuggingFace platform: `https://huggingface.co/models`.

## 6.3 Main findings

In our primary experiments, we focus on LLMs with sizes ranging from 6B to 14B parameters. The outcomes of CR, Acc and SS are presented in Table 1. As previously mentioned, the reported results are the mean value derived from the two conformal score functions, LAC and APS. For a detailed analysis of the results pertaining to each function, please refer to Appendix C.1.

---

[4]We omit Falcon-180B due to insufficient GPU resources.

From Table 1, it is evident that in the majority of cases, the coverage rate is at least 90%, indicating that the coverage guarantee requirement has been met. Although there are cases where the coverage rate falls below 90%,[5] the values are still in close proximity to the 90% threshold. The lowest coverage rate is attained by Qwen-7B on the DS task, with a value of 89.56%. Moreover, all models achieve an average coverage rate exceeding 90% across the five tasks. These findings suggest that the generated prediction sets are meaningful, as they can cover the true label with a high probability. Therefore, the size of the prediction set can serve as a reliable indicator of uncertainty.

In principle, an LLM having higher accuracy is expected to demonstrate lower uncertainty. However, as shown in Table 1, the results regarding the SS metric reveal that in practice, higher accuracy does not necessarily correlate with lower uncertainty. Concretely, for each task, we observe that the ranking of LLMs based on accuracy differs from that based on uncertainty, suggesting that some LLMs possessing higher ac-

Table 1: The evaluation results of LLMs with sizes ranging from 6B to 14B. These results represent the mean values of LAC and APS. The "**Avg.**" column denotes the average performance across the five tasks. The small number in parentheses next to each score indicates the ranking of the LLM for that specific task. The relative ranking of LLMs is also visually demonstrated by the depth of color, with darker colors signifying higher rankings.

| LLMs | QA | RC | CI | DRS | DS | Avg. |
|---|---|---|---|---|---|---|
| *Coverage Rate – CR (%)* | | | | | | |
| Qwen-14B | 92.58 | 95.20 | 95.29 | 91.92 | 89.71 | 92.94 |
| Yi-6B | 91.30 | 94.06 | 92.35 | 91.36 | 91.38 | 92.09 |
| Gemma-7B | 93.57 | 94.16 | 92.13 | 91.59 | 90.70 | 92.43 |
| Mistral-7B | 93.00 | 92.91 | 89.94 | 91.02 | 92.05 | 91.78 |
| Llama-2-13B | 92.59 | 93.15 | 90.50 | 90.92 | 90.82 | 91.60 |
| Qwen-7B | 92.79 | 94.02 | 91.53 | 92.43 | 89.56 | 92.07 |
| InternLM-7B | 90.68 | 93.28 | 90.10 | 90.40 | 90.34 | 90.96 |
| Llama-2-7B | 91.37 | 90.69 | 90.97 | 89.60 | 90.04 | 90.53 |
| DeepSeek-7B | 91.18 | 89.95 | 90.16 | 90.89 | 90.21 | 90.48 |
| MPT-7B | 89.79 | 90.54 | 90.12 | 90.80 | 89.71 | 90.19 |
| Falcon-7B | 90.04 | 89.95 | 89.82 | 90.46 | 90.71 | 90.19 |
| *Prediction Accuracy – Acc (%) ↑* | | | | | | |
| Qwen-14B | 64.25(1) | 91.52(1) | 91.00(1) | 73.90(1) | 49.33(4) | 74.00(1) |
| Yi-6B | 57.57(4) | 85.99(2) | 76.50(2) | 58.72(4) | 66.06(1) | 68.97(2) |
| Gemma-7B | 62.24(2) | 85.29(3) | 73.58(3) | 66.79(2) | 40.80(7) | 65.74(3) |
| Mistral-7B | 60.44(3) | 81.94(5) | 62.93(5) | 53.21(5) | 62.16(2) | 64.14(4) |
| Llama-2-13B | 52.52(6) | 77.23(6) | 59.66(6) | 52.65(6) | 60.05(3) | 60.42(5) |
| Qwen-7B | 55.21(5) | 83.89(4) | 63.70(4) | 64.04(3) | 32.53(9) | 59.87(6) |
| InternLM-7B | 48.37(7) | 73.86(7) | 46.21(7) | 43.72(7) | 34.38(8) | 49.31(7) |
| Llama-2-7B | 45.60(9) | 65.79(8) | 43.05(8) | 32.61(9) | 45.60(5) | 46.53(8) |
| DeepSeek-7B | 45.65(8) | 65.39(9) | 42.66(9) | 33.50(8) | 42.15(6) | 45.87(9) |
| MPT-7B | 29.49(10) | 31.69(10) | 25.50(10) | 24.38(11) | 24.86(10) | 27.18(10) |
| Falcon-7B | 23.75(11) | 24.98(11) | 24.91(11) | 25.86(10) | 24.69(11) | 24.84(11) |
| *Prediction Uncertainty – SS ↓* | | | | | | |
| Qwen-14B | 2.80(2) | 1.74(1) | 2.02(2) | 1.94(1) | 2.37(3) | 2.17(1) |
| Yi-6B | 3.20(5) | 1.92(4) | 1.88(1) | 2.85(6) | 1.96(1) | 2.36(2) |
| Gemma-7B | 2.72(1) | 1.88(3) | 2.04(3) | 2.14(2) | 3.11(7) | 2.38(3) |
| Mistral-7B | 2.80(2) | 1.75(2) | 2.48(5) | 2.71(5) | 2.40(4) | 2.43(4) |
| Llama-2-13B | 3.06(4) | 2.24(7) | 2.72(6) | 2.55(4) | 2.24(2) | 2.56(5) |
| Qwen-7B | 3.26(6) | 2.15(5) | 2.28(4) | 2.51(3) | 2.92(5) | 2.63(6) |
| InternLM-7B | 3.49(9) | 2.19(6) | 3.28(9) | 3.63(10) | 4.47(11) | 3.41(9) |
| Llama-2-7B | 3.20(5) | 2.39(8) | 3.27(8) | 3.26(7) | 3.30(8) | 3.09(7) |
| DeepSeek-7B | 3.34(8) | 2.77(9) | 3.06(7) | 3.40(8) | 3.08(6) | 3.13(8) |
| MPT-7B | 3.53(10) | 3.46(10) | 3.60(10) | 3.59(9) | 3.66(9) | 3.57(10) |
| Falcon-7B | 3.90(11) | 3.60(11) | 3.66(11) | 3.64(11) | 3.92(10) | 3.75(11) |

curacy actually display higher uncertainty. Notably, *two LLMs with a significant difference in accuracy may even display inverse uncertainty*. For example, on the DRS task, InternLM-7B demonstrates higher performance than MPT-7B in accuracy by 19.34 absolute points, yet it shows higher uncertainty. This pattern is also observed on the QA task with Qwen-7B and Llama-2-7B (9.61), the CI task with Qwen-14B and Yi-6B (14.50), and the DS task with InternLM-7B and Falcon-7B (9.69). In each case, the LLM with much higher accuracy exhibits greater uncertainty compared to its counterpart. These observations underscore the importance of considering uncertainty in addition to accuracy when evaluating LLMs.

## 6.4 Effects of model scale

LLMs with larger sizes are usually pretrained on more data and tend to exhibit superior capabilities across various tasks. In this study, we aim to investigate how an LLM's performance changes when scaling its model size. We present the results of the Qwen model series in Figure 3. It is evident that in the majority of cases, the coverage guarantee requirement has been satisfied. In terms of Acc, with the exception of Qwen-72B and Qwen-14B on the CI task, an increase in model size

---

[5]While the theoretical guarantee of conformal prediction is rigorous, there can be minor fluctuations in practice due to finite-sample variability [5].

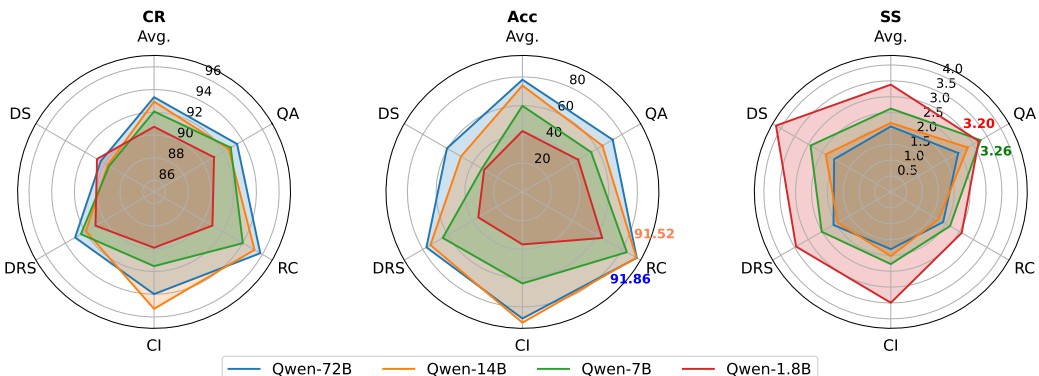

Figure 3: Performance comparison of different versions of the Qwen series (1.8B to 72B).

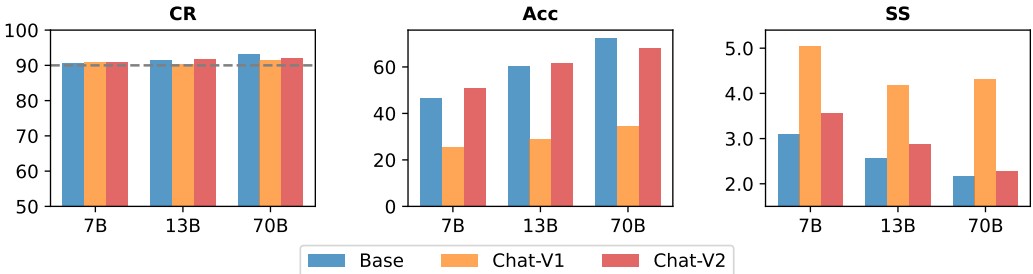

Figure 4: Mean performance outcomes of the **Llama-2** series' base pretrained model and the instruction-finetuned chat model across five tasks. Chat-V1 converts inputs into a chat format. Chat-V2 shares the same format as Base.

consistently leads to improved performance. Concerning uncertainty (i.e. SS), there is a general trend of decreasing uncertainty when scaling the model size from 1.8B to 14B. However, Qwen-7B displays higher uncertainty compared to Qwen-1.8B on the QA task. When further scaling the model size from 14B to 72B, the enhancements in uncertainty become less pronounced, and more variations are observed. Notably, on both the RC and DRS tasks, Qwen-72B demonstrates higher uncertainty than Qwen-14B, although Qwen-72B achieves higher accuracy.

We provide the results of the Llama-2 series, Yi series, DeepSeek series, and Falcon series in Appendix C.2, which reveal similar findings.

### 6.5 Effects of instruction finetuning

In this part, we further delve into the comparative analysis of the performance between the base pretrained model and the instruction-finetuned variant of LLMs. The average results across five tasks of the Llama-2 model series are illustrated in Figure 4. For the instruction-finetuned version, two methods are adopted to prepare the prompt input. The first method aligns with the format of the instruction data[6] (denoted as **Chat-V1**). This method aims to evaluate the model's proficiency in adhering to instructions to accomplish tasks. The second method employs the same prompt format as the base version (denoted as **Chat-V2**). This method aims to assess the extent of the base model's capabilities retained after instruction-finetuning.

Figure 4 shows that Chat-V1 consistently results in lower accuracy and higher uncertainty across all model sizes. Conversely, Chat-V2 enhances accuracy for Llama-2-7B and Llama-2-13B. However, for Llama-2-70B, Chat-V2 also leads to a decline in accuracy. Regarding uncertainty, Chat-V2 consistently results in increased uncertainty compared to the base model, although the extent of degradation is less severe than Chat-V1. These findings suggest that instruction-finetuning tends to impair model performance, particularly in terms of uncertainty.

We provide the results of the Yi series, DeepSeek series, and Falcon series in Appendix C.3.

---

[6]We achieve this by applying the "*apply_chat_template*" function of the corresponding tokenizer to each prompt input.

Table 2: Comparison between conformal prediction (**CP**) and perplexity (**PPL**) using InternLM-7B.

| Tasks | CR (%) | | SS | |
|-------|--------|--------|------|------|
| | **CP** | **PPL** | **CP** | **PPL** |
| QA | 90.68 | 83.44 | 3.49 | 2.89 |
| RC | 93.28 | 95.48 | 2.19 | 2.39 |
| CI | 90.10 | 96.25 | 3.28 | 3.97 |
| DRS | 90.40 | 86.80 | 3.63 | 3.42 |
| DS | 90.34 | 87.13 | 4.47 | 4.33 |
| Avg. | 90.96 | 89.82 | 3.41 | 3.40 |

Table 3: Comparison among conformal prediction (**CP**), entropy (**Entropy**), and maximal predicted probability ($\mathbf{P_{max}}$) using InternLM-7B in terms of **ECE (%)**.

| Tasks | CP | Entropy | $\mathbf{P_{max}}$ |
|-------|-------|---------|-----------|
| QA | 15.83 | 15.83 | 15.83 |
| RC | 1.33 | 1.32 | 1.41 |
| CI | 3.16 | 3.45 | 3.75 |
| DRS | 12.11 | 12.40 | 12.45 |
| DS | 9.30 | 9.30 | 9.62 |
| Avg. | **8.35** | 8.46 | 8.61 |

## 6.6 Comparison to other uncertainty quantification methods

Given the predicted probabilities, another widely used measure of uncertainty is entropy [56]. There have also been some entropy-based uncertainty quantification methods for language models [20]. Here, we compare conformal prediction to entropy. Since conformal prediction uses the prediction set size (SS) to measure uncertainty, a direct comparison with entropy is not straightforward. To address this issue, we convert entropy to perplexity [33], which is defined as $PPL = 2^H$, where $H$ denotes the entropy. Perplexity takes values in the range of $[1, |\mathcal{Y}|]$, allowing it to be interpreted as prediction set size. For instance, when the predicted probability for each class (option) is $\frac{1}{|\mathcal{Y}|}$, $PPL = |\mathcal{Y}|$.

The results regarding InternLM-7B are presented in Table 2, from which, we observe that the coverage rate of perplexity varies significantly across different tasks. On the QA task, the coverage rate is only $83.44\%$. In contrast, the coverage rate of conformal prediction consistently exceeds $90\%$. This is because when measuring uncertainty, entropy doesn't take accuracy into account. Entropy remains the same when predicted probabilities are permuted, even though prediction accuracy may differ.

To further demonstrate the superiority of conformal prediction, we conduct additional experiments comparing it with entropy and maximal predicted probability in terms of the Expected Calibration Error (ECE) metric [27]. The results corresponding to InternLM-7B are presented in Table 3. The observation that conformal prediction yields the lowest average ECE score suggests that it offers more reliable uncertainty quantification.

Overall, these results demonstrate the advantages of adopting conformal prediction for uncertainty quantification. We provide more analyses in Appendix C.7.

## 6.7 Expanding benchmarking to closed-source LLMs

In this part, we extend our benchmarking from open-source LLMs to closed-source LLMs. While obtaining the exact output logits of closed-source LLMs is challenging, we can sample multiple answers and then estimate the probability of each choice. We perform an experiment on the MMLU (the QA task) dataset with GPT-3.5 and GPT-4 as the

Table 4: The evaluation results of closed-source LLMs.

| LLMs | CR (%) | Acc (%) | SS |
|------|--------|---------|------|
| GPT-4 | 90.41 | 81.75 | 1.65 |
| GPT-3.5 | 89.98 | 62.99 | 3.05 |
| Qwen-72B (sampling) | 90.54 | 70.29 | 2.43 |
| Qwen-72B (logits) | 93.34 | 73.55 | 2.33 |

closed-source LLMs. To save cost, we only consider the base prompting strategy. Specifically, we first sample 50 answers for each question and calculate the frequency of each option. Then, we apply the softmax function with temperature scaling to prevent zero probabilities. To demonstrate the quality of this approximation, we also report the results of the open-source model Qwen-72B when getting its predictions via sampling and via logits, respectively. The results are shown in Table 4.

It is observed that GPT-4 demonstrates the highest accuracy and the lowest uncertainty. In addition, the average prediction set size (SS) of Qwen-72B (sampling) is relatively close to that of Qwen-72B (logits). For each question, we further calculate the Jensen-Shannon divergence (JSD) between the predictions of Qwen-72B (sampling) and Qwen-72B (logits). The average JSD is 0.05, indicating that

Table 5: The evaluation results of free-form text generation on TriviaQA.

| LLMs | CR (%) | Acc (%) | SS |
|---|---|---|---|
| Qwen-72B | 88.92 | 76.45 | 2.63 |
| Llama-2-13B | 83.89 | 71.83 | 2.40 |
| Qwen-14B | 82.79 | 66.57 | 3.83 |
| Llama-2-7B | 78.83 | 64.91 | 3.06 |
| Qwen-7B | 77.89 | 59.44 | 5.02 |
| DeepSeek-7B | 78.41 | 57.52 | 6.12 |
| Falcon-7B | 76.51 | 55.74 | 6.27 |

Table 6: The relationship between stratified prediction set size (SS) and prediction accuracy (Acc).

| SS | LAC | APS | Avg. |
|---|---|---|---|
| 1 | 80.39 | 92.69 | 86.54 |
| 2 | 59.77 | 82.21 | 70.99 |
| 3 | 40.55 | 63.70 | 52.12 |
| 4 | 40.07 | 41.71 | 40.89 |
| 5 | 31.50 | 34.92 | 33.21 |
| 6 | 13.43 | None | 13.42 |

the two predictions (estimated probability distributions) are highly similar. Therefore, we conclude that the approximation is of high quality.

## 6.8 Expanding benchmarking to free-form text generation

Here, we further extend our benchmarking from multiple-choice question answering to free-form text generation. However, applying conformal prediction to text generation is a complex task due to the extensive range of potential responses. It is not feasible to compute the probability for each possible response and then use conformal prediction to select a subset. Nevertheless, many potential responses have a low probability of being generated, which allows us to reduce the selection space by sampling multiple generations.

Specifically, we adopt the TriviaQA dataset [35] (sampling 10,000 dev instances) for free-form text generation. We first generate 20 answers for each question. Then, we employ the perplexity [33] of each generation as the conformal score function and utilize exact match to verify the accuracy of the generated answer. The results are displayed in Table 5. Note that the value of SS falls into $[1, 20]$.

From Table 5, we observe that the prediction set size (SS) varies among LLMs, which could provide some insights into the uncertainty of these models. However, we must note that in this sampling setting, the coverage rate cannot be guaranteed any more because there might not be a correct answer within the 20 sampled responses. In other words, even if the prediction set size is 20, indicating high model uncertainty, the coverage rate for that instance could still be zero if there are no correct answers present. Nonetheless, it is observed that when the LLM is stronger, the coverage guarantee requirement is more likely to be satisfied.

## 6.9 In-depth analysis of the set size metric in relation to prediction accuracy

While our main focus is on the high probability of the prediction set covering the ground truth, it is also insightful to explore the relationship between stratified set size and prediction accuracy. In our experiments, we employ InternLM-7B on the QA task, grouping instances by their predicted set size and reporting the accuracy within each group in Table 6. The results reveal that instances with smaller set sizes are generally associated with higher prediction accuracy, indicating that set size serves as a useful indicator of prediction uncertainty. However, it is important to note that even when the set size reaches its maximum value, there are still instances where the prediction is accurate. Consequently, a comprehensive analysis of both prediction accuracy and prediction uncertainty is essential for a thorough assessment of the performance of LLMs.

## 7 Conclusion

In this work, we have provided an extensive examination of the performance of LLMs by focusing on prediction uncertainty. To achieve this, we have employed conformal prediction for uncertainty quantification. Our comprehensive investigation, which involves nine open-source LLMs (or LLM series) and spans five typical NLP tasks, demonstrates that relying solely on accuracy for benchmarking LLMs is insufficient. Instead, it is imperative to take uncertainty into account when assessing their overall performance. Last but not least, we have verified the superiority of conformal prediction compared to several other uncertainty quantification methods. We have also extended our analyses to closed-source LLMs and free-form text generation.

## Acknowledgments and Disclosure of Funding

This work was supported in part by the Tencent AI Lab Rhino-Bird (Grant No. EF2023-00151-FST), the Science and Technology Development Fund, Macau SAR (Grant Nos. FDCT/060/2022/AFJ, FDCT/0070/2022/AMJ), and the Multi-year Research Grant from the University of Macau (Grant No. MYRG-GRG2023-00006-FST-UMDF). We thank all reviewers for their precious comments.

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

# A    Pseudo code

---

**Algorithm 1** Conformal Prediction for Uncertainty Quantification of LLMs

---

**Require:** An LLM $\mathcal{M}$, a task-specific dataset $\mathcal{D}$, a user-specified error rate $\alpha$, a test data ratio $\beta$, a prompting
strategy $\mathcal{P}$, and a conformal score function $s$ (either LAC or APS);
**Output:** Evaluation results in terms of evaluation metrics $Acc$, $SS$, and $CR$;
1:  ▷ **Get model predictions**
2:  $\mathcal{O} \leftarrow$ Initialized as an empty list;
3:  **for each** instance $X$ in $\mathcal{D}$ **do**
4:      $X' \leftarrow$ FormatPromptInput($X$, $\mathcal{P}$);
5:      $L(X) \leftarrow$ GetLogitsOfOptions($\mathcal{M}$, $X'$);
6:      $P(X) \leftarrow$ Softmax($L(X)$);
7:      Append $P(X)$ to $\mathcal{O}$;
8:  **end for**
9:  $\mathcal{D}_{cal}, \mathcal{D}_{test} \leftarrow$ CalibrationTestSplit($\mathcal{D}$, $\beta$);
10: $\mathcal{O}_{cal}, \mathcal{O}_{test} \leftarrow$ CalibrationTestSplit($\mathcal{O}$, $\beta$);
11: $Acc \leftarrow$ CalculateAccuracy($\mathcal{D}_{test}$, $\mathcal{O}_{test}$);
12: ▷ **Apply conformal prediction**
13: $\mathcal{S} \leftarrow$ ComputeConformalScores($\mathcal{D}_{cal}$, $\mathcal{O}_{cal}$, $s$);
14: $\hat{p} \leftarrow$ CalculateConformalThreshold($\mathcal{S}$, $\alpha$);
15: $\mathcal{B} \leftarrow$ Initialized as an empty list;
16: **for each** instance $X$ in $\mathcal{D}_{test}$ **do**
17:     Get $P(X)$ from $\mathcal{O}_{test}$;
18:     $\mathcal{C}(X) \leftarrow$ CreatePredictionSet($P(X)$, $\hat{p}$, $s$);
19:     **if** $|\mathcal{C}(X)| == 0$ **then**
20:         $\mathcal{C}(X) \leftarrow$ {Option with the largest probability};
21:     **end if**
22:     Append $C(X)$ to $\mathcal{B}$;
23: **end for**
24: $SS \leftarrow$ CalculateAverageSetSize($\mathcal{B}$);
25: $CR \leftarrow$ CalculateCoverageRate($\mathcal{B}$, $\mathcal{D}_{test}$);
26: **return** $Acc$, $SS$, $CR$.

---

We present a summary of the pseudo code for applying conformal prediction to quantify the uncertainty of LLMs in Algorithm 1. The procedure is outlined as follows:

1. For each instance, input it into the LLM to obtain the logits output for all possible options.

2. Apply the softmax function to transform these logits into probability values.

3. Divide the dataset into a calibration set and a test set.

4. Employ the user-specified error rate $\alpha$ and the calibration set to determine the conformal threshold.

5. Generate prediction sets for instances in the test set based on the conformal threshold.

6. In the event that a prediction set is empty, select the option with the highest probability as the final prediction.

7. Calculate the evaluation metrics, namely Acc, SS, and CR.

In our experiments, we use a server with eight A100 40GB cards to load each LLM checkpoint and perform inference with a batch size of 1.

# B    Dataset statistics

Figure 5 presents the distribution of correct answer choices for each task. It is noteworthy that while we have incorporated options E ("*I don't know*") and F ("*None of the above*") for every question, the correct answer consistently falls within the set {A, B, C, D}. As depicted in Figure 5, the distribution of correct answers is nearly uniform across options A, B, C, and D for all tasks except the QA task. However, even on the QA task, the distribution does not exhibit a significant skew. These statistics indicate that the created datasets are suitable for rigorously evaluating the performance of LLMs.

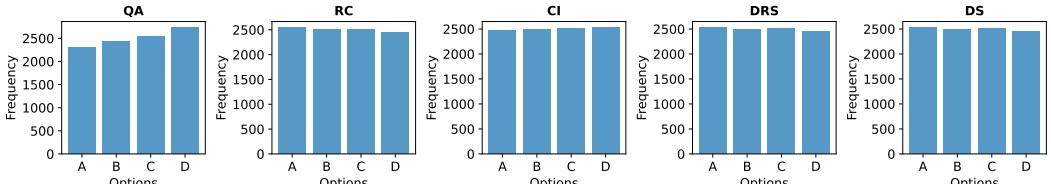

Figure 5: The distributions of correct answer choices on each task.

Table 7: The CR (%) results of LLMs with sizes ranging from 6B to 14B. The "**Avg.**" column denotes the average performance across tasks. These results correspond to using LAC and APS as the conformal score function separately.

| LLMs | LAC | | | | | | APS | | | | | |
|---|---|---|---|---|---|---|---|---|---|---|---|---|
| | QA | RC | CI | DRS | DS | Avg. | QA | RC | CI | DRS | DS | Avg. |
| Qwen-14B | 90.43 | 91.57 | 91.40 | 90.31 | 89.63 | 90.67 | 94.74 | 98.83 | 99.19 | 93.52 | 89.79 | 95.21 |
| Yi-6B | 89.96 | 90.04 | 89.83 | 90.96 | 90.40 | 90.24 | 92.64 | 98.09 | 94.86 | 91.77 | 92.37 | 93.95 |
| Gemma-7B | 90.38 | 90.48 | 90.57 | 90.30 | 90.81 | 90.51 | 96.76 | 97.85 | 93.70 | 92.87 | 90.60 | 94.35 |
| Mistral-7B | 89.64 | 89.48 | 89.99 | 91.01 | 90.86 | 90.20 | 96.37 | 96.33 | 89.88 | 91.04 | 93.24 | 93.37 |
| Llama-2-13B | 90.06 | 89.91 | 90.32 | 90.60 | 90.46 | 90.27 | 95.12 | 96.39 | 90.68 | 91.25 | 91.18 | 92.92 |
| Qwen-7B | 90.54 | 89.80 | 89.88 | 90.50 | 90.00 | 90.14 | 95.04 | 98.25 | 93.18 | 94.35 | 89.12 | 93.99 |
| InternLM-7B | 89.38 | 90.33 | 89.96 | 90.61 | 90.62 | 90.18 | 91.98 | 96.23 | 90.24 | 90.18 | 90.06 | 91.74 |
| Llama-2-7B | 90.42 | 89.61 | 91.00 | 89.79 | 89.85 | 90.13 | 92.33 | 91.76 | 90.95 | 89.40 | 90.24 | 90.94 |
| DeepSeek-7B | 89.94 | 88.48 | 89.79 | 91.09 | 90.16 | 89.89 | 92.42 | 91.43 | 90.53 | 90.70 | 90.26 | 91.07 |
| MPT-7B | 90.24 | 90.77 | 90.19 | 90.89 | 89.48 | 90.31 | 89.35 | 90.32 | 90.06 | 90.72 | 89.94 | 90.08 |
| Falcon-7B | 90.06 | 89.94 | 89.77 | 90.72 | 90.60 | 90.22 | 90.01 | 89.96 | 89.86 | 90.20 | 90.82 | 90.17 |

Table 8: The Acc and SS results of LLMs with sizes ranging from 6B to 14B. These results are obtained when LAC is adopted as the conformal score function. The "**Avg.**" column denotes the average performance across tasks. The small number in parentheses indicates the rank of the model on each task.

| LLMs | Acc (%) ↑ | | | | | | SS ↓ | | | | | |
|---|---|---|---|---|---|---|---|---|---|---|---|---|
| | QA | RC | CI | DRS | DS | Avg. | QA | RC | CI | DRS | DS | Avg. |
| Qwen-14B | 64.25(1) | 91.52(1) | 91.00(1) | 73.90(1) | 49.33(4) | 74.00(1) | 2.39(3) | 1.00(1) | 1.01(1) | 1.54(1) | 2.26(4) | 1.64(1) |
| Yi-6B | 57.57(4) | 85.99(2) | 76.50(2) | 58.72(4) | 66.06(1) | 68.97(2) | 2.92(6) | 1.12(2) | 1.53(2) | 2.74(6) | 1.60(1) | 1.98(2) |
| Gemma-7B | 62.24(2) | 85.29(3) | 73.58(3) | 66.79(2) | 40.80(7) | 65.74(3) | 2.23(1) | 1.18(3) | 1.79(3) | 1.97(2) | 3.17(7) | 2.07(3) |
| Mistral-7B | 60.44(3) | 81.94(5) | 62.93(5) | 53.21(5) | 62.16(2) | 64.14(4) | 2.32(2) | 1.26(5) | 2.36(5) | 2.62(5) | 2.14(3) | 2.14(4) |
| Llama-2-13B | 52.52(6) | 77.23(6) | 59.66(6) | 52.65(6) | 60.05(3) | 60.42(5) | 2.73(4) | 1.58(6) | 2.58(6) | 2.53(4) | 2.12(2) | 2.31(6) |
| Qwen-7B | 55.21(5) | 83.89(4) | 63.70(4) | 64.04(3) | 32.53(9) | 59.87(6) | 2.82(5) | 1.21(4) | 2.02(4) | 2.08(3) | 2.93(5) | 2.21(5) |
| InternLM-7B | 48.37(7) | 73.86(7) | 46.21(7) | 43.72(7) | 34.38(8) | 49.31(7) | 3.23(9) | 1.71(7) | 3.17(8) | 3.54(9) | 4.43(11) | 3.22(9) |
| Llama-2-7B | 45.60(9) | 65.79(8) | 43.05(8) | 32.61(9) | 45.60(5) | 46.53(8) | 3.05(7) | 2.20(8) | 3.23(9) | 3.25(7) | 3.45(8) | 3.03(7) |
| DeepSeek-7B | 45.65(8) | 65.39(9) | 42.66(9) | 33.50(8) | 42.15(6) | 45.87(9) | 3.19(8) | 2.50(9) | 3.01(7) | 3.38(8) | 3.09(6) | 3.03(7) |
| MPT-7B | 29.49(10) | 31.69(10) | 25.50(10) | 24.38(11) | 24.86(10) | 27.18(10) | 3.54(10) | 3.44(10) | 3.60(10) | 3.62(10) | 3.63(9) | 3.57(10) |
| Falcon-7B | 23.75(11) | 24.98(11) | 24.91(11) | 25.86(10) | 24.69(11) | 24.84(11) | 3.92(11) | 3.59(11) | 3.64(11) | 3.66(11) | 3.93(10) | 3.75(11) |

# C  Further experimental results

## C.1  Detailed results of LAC and APS

Table 1 has presented the average results derived from the two conformal score functions, namely, LAC and APS. In this part, we analyze the results associated with each conformal score function. The detailed results are reported in Table 7, Table 8, and Table 9.

It is evident that for both conformal score functions, the ranking of LLMs based on accuracy can be different from that based on uncertainty. These results reaffirm the importance of considering uncertainty in order to evaluate the performance of LLMs in a more holistic manner. Another notable observation is the difference in the uncertainty estimations produced by LAC and APS. In general, APS tends to produce larger prediction sets. More importantly, APS can lead to a significantly different ranking of LLMs based on uncertainty compared to LAC. For example, on the CI task, Qwen-14B secures the lowest uncertainty when LAC is utilized as the conformal score function. However, when APS is employed as the conformal score function, Qwen-14B is ranked sixth. This observation suggests that it is essential to average the results of the two conformal score functions

Table 9: The Acc and SS results of LLMs with sizes ranging from 6B to 14B. These results are obtained when APS is adopted as the conformal score function. The "**Avg.**" column denotes the average performance across tasks. The small number in parentheses indicates the rank of the model on each task.

| LLMs | Acc (%) ↑ | | | | | | SS ↓ | | | | | |
|---|---|---|---|---|---|---|---|---|---|---|---|---|
| | QA | RC | CI | DRS | DS | Avg. | QA | RC | CI | DRS | DS | Avg. |
| Qwen-14B | $64.25_{(1)}$ | $91.52_{(1)}$ | $91.00_{(1)}$ | $73.90_{(1)}$ | $49.33_{(4)}$ | $74.00_{(1)}$ | $3.21_{(1)}$ | $2.47_{(2)}$ | $3.03_{(6)}$ | $2.33_{(2)}$ | $2.47_{(3)}$ | $2.70_{(2)}$ |
| Yi-6B | $57.57_{(4)}$ | $85.99_{(2)}$ | $76.50_{(2)}$ | $58.72_{(4)}$ | $66.06_{(1)}$ | $68.97_{(2)}$ | $3.48_{(6)}$ | $2.72_{(6)}$ | $2.22_{(1)}$ | $2.95_{(5)}$ | $2.33_{(1)}$ | $2.74_{(4)}$ |
| Gemma-7B | $62.24_{(2)}$ | $85.29_{(3)}$ | $73.58_{(3)}$ | $66.79_{(2)}$ | $40.80_{(7)}$ | $65.74_{(3)}$ | $3.21_{(1)}$ | $2.57_{(3)}$ | $2.29_{(2)}$ | $2.31_{(1)}$ | $3.05_{(6)}$ | $2.68_{(1)}$ |
| Mistral-7B | $60.44_{(3)}$ | $81.94_{(5)}$ | $62.93_{(5)}$ | $53.21_{(5)}$ | $62.16_{(2)}$ | $64.14_{(4)}$ | $3.27_{(3)}$ | $2.25_{(1)}$ | $2.59_{(4)}$ | $2.80_{(4)}$ | $2.67_{(4)}$ | $2.71_{(3)}$ |
| Llama-2-13B | $52.52_{(6)}$ | $77.23_{(6)}$ | $59.66_{(6)}$ | $52.65_{(6)}$ | $60.05_{(3)}$ | $60.42_{(5)}$ | $3.40_{(5)}$ | $2.90_{(7)}$ | $2.86_{(5)}$ | $2.58_{(3)}$ | $2.36_{(2)}$ | $2.82_{(5)}$ |
| Qwen-7B | $55.21_{(5)}$ | $83.89_{(4)}$ | $63.70_{(4)}$ | $64.04_{(3)}$ | $32.53_{(9)}$ | $59.87_{(6)}$ | $3.70_{(9)}$ | $3.10_{(9)}$ | $2.53_{(3)}$ | $2.95_{(5)}$ | $2.91_{(5)}$ | $3.04_{(6)}$ |
| InternLM-7B | $48.37_{(7)}$ | $73.86_{(7)}$ | $46.21_{(7)}$ | $43.72_{(7)}$ | $34.38_{(8)}$ | $49.31_{(7)}$ | $3.74_{(10)}$ | $2.68_{(5)}$ | $3.39_{(9)}$ | $3.71_{(11)}$ | $4.51_{(11)}$ | $3.61_{(10)}$ |
| Llama-2-7B | $45.60_{(9)}$ | $65.79_{(8)}$ | $43.05_{(8)}$ | $32.61_{(9)}$ | $45.60_{(5)}$ | $46.53_{(8)}$ | $3.35_{(4)}$ | $2.58_{(4)}$ | $3.32_{(8)}$ | $3.27_{(7)}$ | $3.15_{(8)}$ | $3.14_{(7)}$ |
| DeepSeek-7B | $45.65_{(8)}$ | $65.39_{(9)}$ | $42.66_{(9)}$ | $33.50_{(8)}$ | $42.15_{(6)}$ | $45.87_{(9)}$ | $3.48_{(6)}$ | $3.03_{(8)}$ | $3.12_{(7)}$ | $3.42_{(8)}$ | $3.07_{(7)}$ | $3.23_{(8)}$ |
| MPT-7B | $29.49_{(10)}$ | $31.69_{(10)}$ | $25.50_{(10)}$ | $24.38_{(11)}$ | $24.86_{(10)}$ | $27.18_{(10)}$ | $3.53_{(8)}$ | $3.49_{(10)}$ | $3.60_{(10)}$ | $3.55_{(9)}$ | $3.69_{(9)}$ | $3.57_{(9)}$ |
| Falcon-7B | $23.75_{(11)}$ | $24.98_{(11)}$ | $24.91_{(11)}$ | $25.86_{(10)}$ | $24.69_{(11)}$ | $24.84_{(11)}$ | $3.89_{(11)}$ | $3.60_{(11)}$ | $3.69_{(11)}$ | $3.62_{(10)}$ | $3.91_{(10)}$ | $3.74_{(11)}$ |

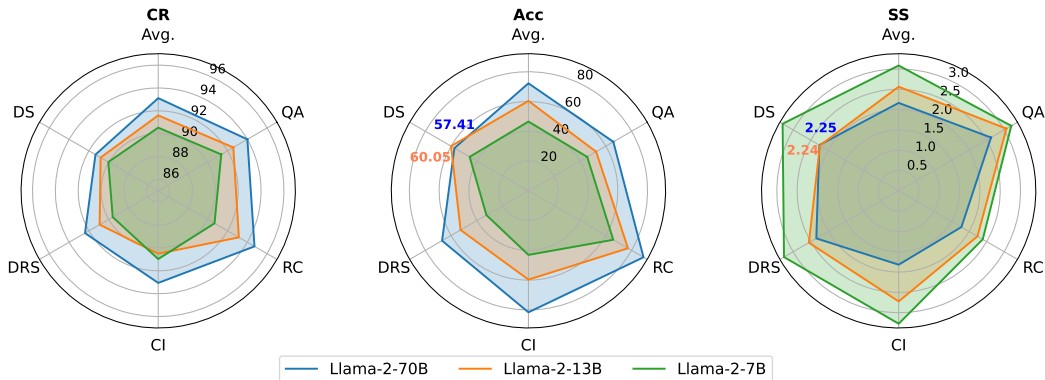

Figure 6: Performance comparison of different versions of the Llama-2 series (7B to 70B).

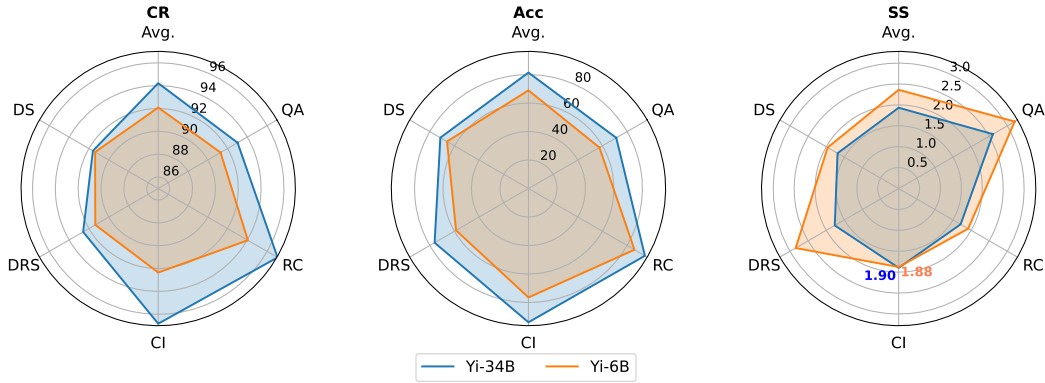

Figure 7: Performance comparison of different versions of the Yi series (6B to 34B).

to provide a more accurate quantification of uncertainty. Last but not least, both conformal score functions can achieve high coverage rates, verifying again the rationality of relying on prediction set size to estimate uncertainty. It is also noted that APS tends to achieve higher coverage rates than LAC due to its larger prediction sets in most cases.

## C.2 Effects of model scale (cont.)

Figures 6-9 illustrate the performance outcomes of the Llama-2 series, Yi series, DeepSeek series, and Falcon series. It is observed that while in general, increasing model size can lead to stronger performance, on some tasks, a larger model may display weaker performance. For example, on the

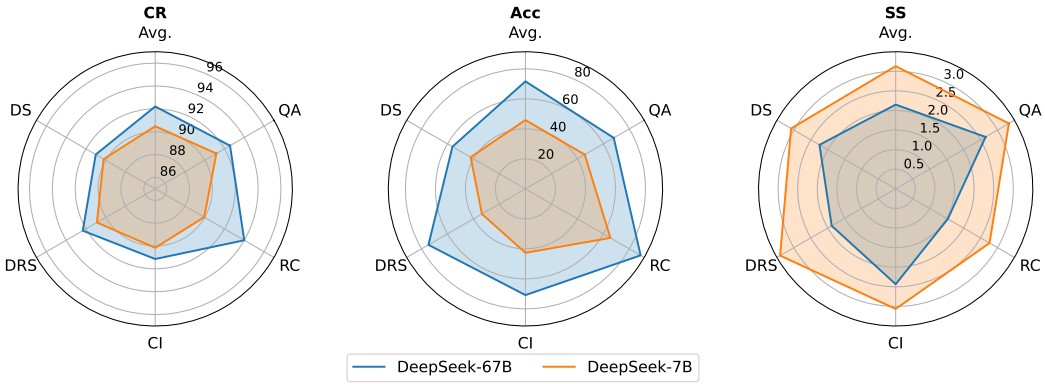

Figure 8: Performance comparison of different versions of the DeepSeek series (7B to 67B).

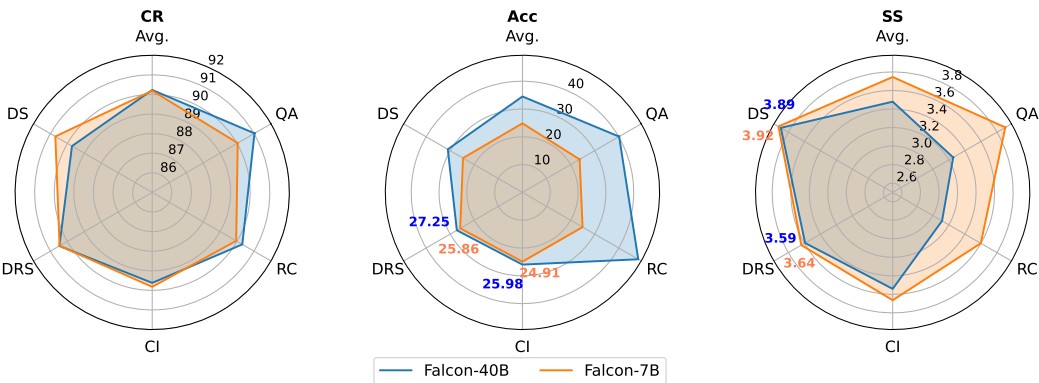

Figure 9: Performance comparison of different versions of the Falcon series (7B to 40B).

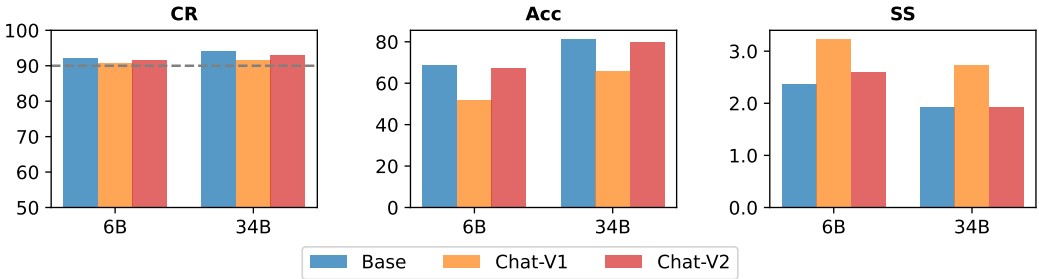

Figure 10: Mean performance outcomes of the **Yi** series' base pretrained model and the instruction-finetuned chat model across five tasks. Chat-V1 converts inputs into a chat format. Chat-V2 shares the same format as Base.

DS task, Llama-2-70B exhibits inferior performance compared to Llama-2-13B across both accuracy and uncertainty. On the CI task, although Yi-34B demonstrates higher accuracy than Yi-6B, it shows higher uncertainty.

## C.3 Effects of instruction finetuning (cont.)

Figures 10-12 depict the average results across five tasks of the Yi series, DeepSeek series, and Falcon series, as well as their instruction-finetuned counterparts. Recall that for the instruction-finetuned version, two approaches are adopted to prepare the prompt input. The first approach adheres to the format of the instruction data, which is denoted as **Chat-V1**. The second approach employs the same prompt format as the base version and is denoted as **Chat-V2**. For the Yi series, it is observed that

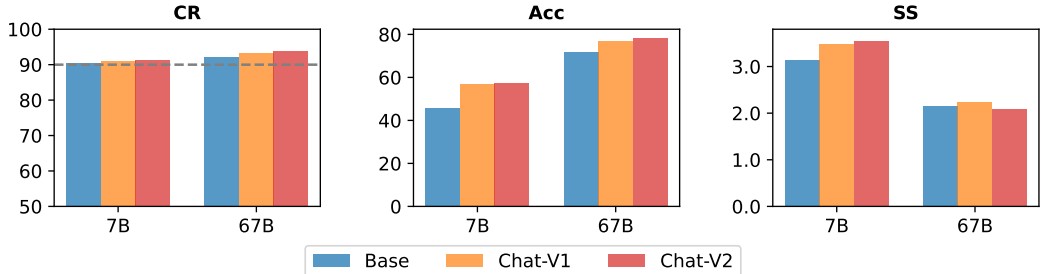

Figure 11: Mean performance outcomes of the **DeepSeek** series' base pretrained model and the instruction-finetuned chat model across five tasks. Chat-V1 converts inputs into a chat format. Chat-V2 shares the same format as Base.

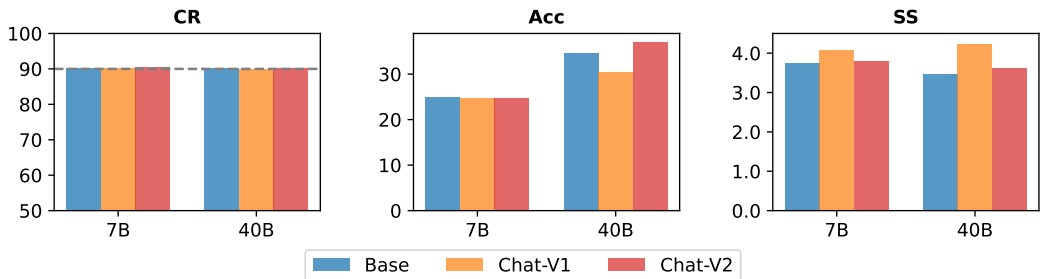

Figure 12: Mean performance outcomes of the **Falcon** series' base pretrained model and the instruction-finetuned chat model across five tasks. Chat-V1 converts inputs into a chat format. Chat-V2 shares the same format as Base.

both Chat-V1 and Chat-V2 consistently yield inferior performance than the base model in terms of both Acc and SS. In contrast, for the DeepSeek series, both Chat-V1 and Chat-V2 exhibit enhanced performance in terms of Acc. Nevertheless, Chat-V1 results in greater uncertainty compared to the base model for both DeepSeek-7B and DeepSeek-70B. Chat-V2 also demonstrates higher uncertainty relative to the base model for DeepSeek-7B. For the Falcon series, it is observed that both Chat-V1 and Chat-V2 consistently lead to higher uncertainty than the base model, even though Chat-V2 achieves better performance in terms of Acc.

### C.4 Effects of mixture of experts

A Mixture of Experts (MoE) is a technique that combines multiple specialized models with a gating mechanism to enhance performance by leveraging diverse expertise and adaptability in handling complex data relationships. In recent months, the adoption of the MoE technique to augment the performance of LLMs has been steadily increasing. Considering this, we study how MoE impacts the uncertainty of LLMs, specifically comparing the Mixtral-8x7B MoE model[7] with Mistral-7B. The results are reported in Table 10. While Mixtral-8x7B has 46.7B total parameters, it only uses 12.9B parameters per token. It, therefore, processes input and generates output at the same speed and for the same cost as a 12.9B model.[8] To provide a comprehensive comparison, we also include the results of Llama-2-13B and Qwen-14B. As can be observed, Mixtral-8x7B consistently outperforms Mistral-7B across both accuracy and uncertainty. Remarkably, it achieves accuracy levels comparable to Qwen-14B, while demonstrating the lowest

Table 10: Average results across five tasks of Mixtral-8x7B, Mistral-7B, Llama-2-13B, and Qwen-14B.

| LLMs | CR (%) | Acc (%) ↑ | SS ↓ |
|------|--------|-----------|------|
| Mixtral-8x7B | 92.59 | 73.74 | 2.03 |
| Mistral-7B | 91.78 | 64.14 | 2.43 |
| Llama-2-13B | 91.60 | 60.42 | 2.56 |
| Qwen-14B | 92.94 | 74.00 | 2.17 |

---

[7]https://huggingface.co/mistralai/Mixtral-8x7B-v0.1
[8]https://mistral.ai/news/mixtral-of-experts/

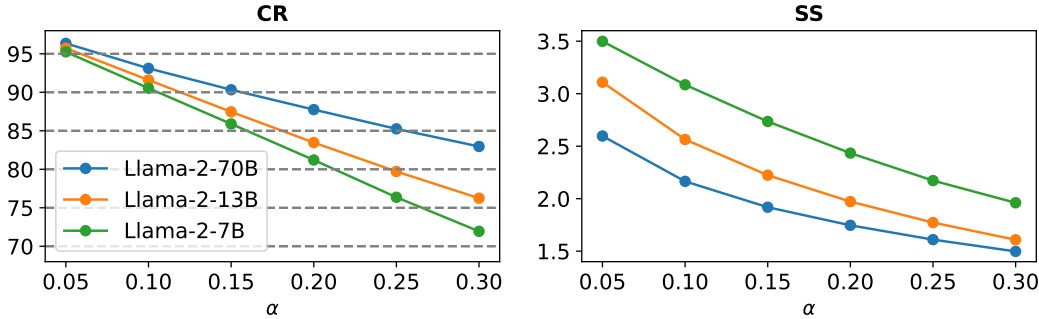

Figure 13: Average results across five tasks of the Llama-2 series when varying the error rate $\alpha$. Note that the ideal coverage rate should be no less than $1 - \alpha$.

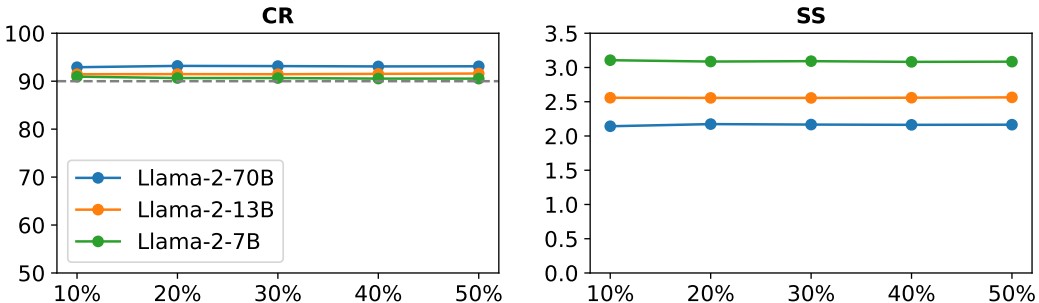

Figure 14: Average results across five tasks of the Llama-2 series when varying the proportion of calibration data.

average uncertainty among the four LLMs. These observations show that MoE is indeed an effective method to enhance the accuracy and reduce the uncertainty of LLMs.

### C.5 Effects of error rate

Conformal prediction ensures that the prediction set encompasses the true label with a probability no less than $1 - \alpha$, where $\alpha$ denotes a user-specified error rate. In consideration of this, it is imperative to investigate the impact of varying the value of the error rate $\alpha$ on the prediction set and, subsequently, the estimation of uncertainty. For this purpose, we vary the value of $\alpha$ within the range of 0.05 to 0.3 and report the results of the Llama-2 series in Figure 13. It is observed that as the value of $\alpha$ increases, the coverage rate decreases monotonically. This outcome is anticipated, as a higher error rate implies a reduced probability of the true label being included in the prediction set. Nevertheless, the coverage rate consistently remains greater than $1 - \alpha$. This observation reaffirms the statistical guarantee provided by conformal prediction in generating the prediction set. It is also observed that the average set size (SS) decreases monotonically with an increase in the value of $\alpha$. This observation is logical, as a larger error rate suggests that the prediction set can miss the true label with a higher probability and consequently, have a smaller size. Another noteworthy observation is that, regardless of the value of $\alpha$, Llama-2-70B consistently exhibits lower uncertainty than Llama-2-13B, and Llama-2-13B consistently displays lower uncertainty than Llama-2-7B. This finding is of paramount importance, as it demonstrates that although different values of the error rate $\alpha$ can lead to varying sizes of the prediction set, the relative rankings of different LLMs based on uncertainty remain unchanged.

### C.6 Effects of amount of calibration data

As described in § 3, conformal prediction requires a calibration set to calculate the threshold $\hat{q}$. In our prior analyses, we have allocated 50% of the data as the calibration set and the remaining 50% as the test set. Here, we explore the impact of varying the proportion of calibration data, ranging from 10% to 50%, on uncertainty quantification. Note that the same 50% of data is consistently used as the test

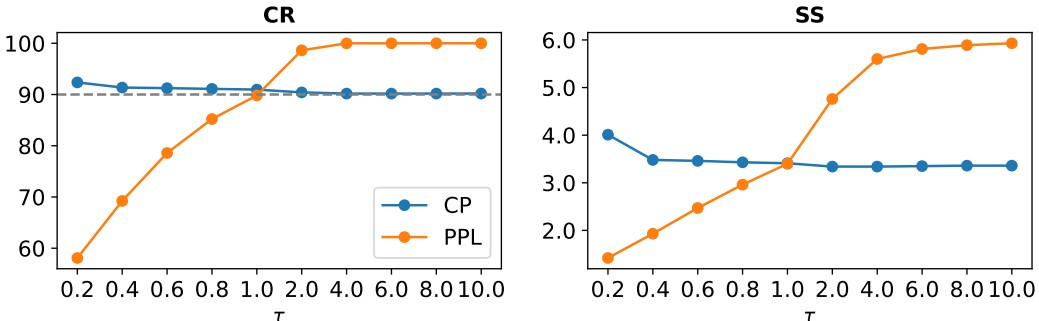

Figure 15: Average results across five tasks of InternLM-7B when varying the softmax temperature $\tau$. We compare the performance of conformal prediction (**CP**) and perplexity (**PPL**).

set. The average results across five tasks of the Llama-2 series are shown in Figure 14. It is observed that there are no significant variations in coverage rate (CR) and uncertainty (SS) for all versions of the Llama-2 series when varying the amount of calibration data. This observation confirms the efficacy of applying conformal prediction for uncertainty quantification in our analysis.

### C.7 Effects of softmax temperature

Recall that we utilize the softmax function to convert option logits generated by LLMs into probability values. These probability values are then employed by the LAC and APS score functions to estimate uncertainty. In practice, we can incorporate a temperature parameter $\tau$ into the softmax function to adjust the probability distributions [11], as shown in the following equation:

$$softmax(z_i, \tau) = \frac{e^{\frac{z_i}{\tau}}}{\sum_{j=1}^{m} e^{\frac{z_j}{\tau}}}, \tag{8}$$

where $z = (z_1, \ldots, z_m) \in \mathbb{R}^m$. A higher temperature results in a more uniform probability distribution (i.e. with higher entropy and "more random"), while a lower temperature leads to a sharper probability distribution, with one value dominating. In this part, we investigate how the temperature $\tau$ affects uncertainty estimation when using conformal prediction and perplexity, respectively. We modify the value of $\tau$ from 0.2 to 10.0 and report the results of InternLM-7B in Figure 15. Note that the temperature does not affect accuracy, so we omit results regarding accuracy and only provide results of CR and SS.

It is observed that when using conformal prediction, we can obtain relatively stable performance in terms of both coverage rate and uncertainty (measured by SS). However, perplexity is highly sensitive to the temperature $\tau$. When $\tau$ takes small values, perplexity results in high certainty but low coverage rate. Note that when we vary the value of $\tau$, we do not change the LLM's accuracy. Thus, a low temperature causes the LLM to be overconfident. In this scenario, it is actually not ideal to estimate low uncertainty. Conformal prediction penalizes this phenomenon by producing large prediction sets (implying high uncertainty), which is more desirable. It is also observed that when $\tau$ takes large values and the probability distribution is close to a uniform distribution (indicating high entropy and that the LLM becomes underconfident), perplexity produces large prediction sets, which are uninformative. In contrast, conformal prediction is able to produce compact prediction sets consistently. This advantage is attributed to the use of a calibration set in conformal prediction. In summary, these observations suggest that conformal prediction is a more reliable method than entropy or perplexity for evaluating uncertainty.

### C.8 Unify all tasks as one

In our previous analyses, we treat each task as an independent one. Therefore, we need to calculate a separate conformal threshold for each task and generate the prediction set based on the task-specific threshold. However, considering that LLMs are capable of solving multiple tasks, it is possible to consider all five tasks in this study as a single unified task (assuming that the datasets corresponding

Table 11: Results of unifying all tasks as a single joint one for which a common conformal threshold is computed for all tasks and the prediction sets are generated based on this shared threshold (reported in the "**Joint**" column). For the sake of comparison, we also include the average results of the five tasks when treated individually (reported in the "**Average**" column). Note that both settings achieve the same performance in terms of accuracy.

| LLMs | Acc (%) ↑ | CR (%) | | SS ↓ | |
|------|-----------|--------|------|------|------|
| | | **Average** | **Joint** | **Average** | **Joint** |
| Yi-34B | 81.46 | 94.22 | 94.47 | 1.93 | 2.18 |
| Qwen-72B | 78.05 | 93.33 | 92.29 | 2.06 | 2.10 |
| Qwen-14B | 74.00 | 92.94 | 94.04 | 2.17 | 2.39 |
| Llama-2-70B | 72.24 | 93.11 | 92.96 | 2.16 | 2.23 |
| DeepSeek-67B | 71.66 | 92.21 | 91.75 | 2.15 | 2.29 |
| Yi-6B | 68.97 | 92.09 | 92.77 | 2.36 | 2.49 |
| Gemma-7B | 65.74 | 92.43 | 92.21 | 2.38 | 2.49 |
| Mistral-7B | 64.14 | 91.78 | 92.05 | 2.43 | 2.47 |
| Llama-2-13B | 60.42 | 91.60 | 91.97 | 2.56 | 2.65 |
| Qwen-7B | 59.87 | 92.07 | 92.70 | 2.63 | 2.69 |
| InternLM-7B | 49.31 | 90.96 | 91.08 | 3.41 | 3.45 |
| Llama-2-7B | 46.53 | 90.53 | 90.53 | 3.09 | 3.15 |
| DeepSeek-7B | 45.87 | 90.48 | 90.59 | 3.13 | 3.18 |
| Qwen-1.8B | 42.34 | 90.73 | 90.60 | 3.38 | 3.39 |
| Falcon-40B | 34.50 | 90.23 | 90.15 | 3.48 | 3.49 |
| MPT-7B | 27.18 | 90.19 | 90.26 | 3.57 | 3.61 |
| Falcon-7B | 24.84 | 90.19 | 90.25 | 3.75 | 3.81 |

to the five tasks are drawn from a joint distribution). By doing so, we only need to calculate one conformal threshold and generate the prediction set for all tasks based on this shared threshold. In particular, we combine the calibration sets of all tasks into one and similarly merge the test sets of all tasks into one. The results of various LLMs using this unified approach are presented in Table 11, where we also include the average results of the five tasks when treated individually for comparison purposes. It can be observed that when treating all tasks as a single (joint) one, all LLMs are still able to meet the coverage guarantee requirement. However, they exhibit higher uncertainty in terms of the average set size (SS). This finding suggests that while LLMs are indeed capable of addressing multiple tasks, it remains crucial to analyze each task independently since LLMs can demonstrate varying degrees of uncertainty across different tasks.

### C.9 Rate of predicted options being E or F

When preparing datasets, in order to enhance the complexity of tasks and effectively quantify uncertainty, two additional answer choices, E ("I don't know") and F ("None of the above"), are incorporated into the datasets for each task. It is important to note that neither of these options represents the correct answer for any of the questions. With this in mind, our study aims to investigate whether an LLM might predict options E or F as the answer, and if so, the number of test instances for which such predictions would be made. Table 12 presents the results of various LLMs, demonstrating that, across all LLMs, only a tiny proportion of test instances are predicted to have a true answer of either E or F. Furthermore, we observe that for a particular LLM, there can be no questions whose predicted answer is E or F on some tasks (e.g., Mistral-7B on the RC task).

We also report the average prediction set size (SS) when options E and F are excluded from the answer choices to evaluate their impact on uncertainty quantification. The results, presented in Table 13, indicate that the SS values remain relatively consistent with those observed when options E and F are included. Furthermore, while smaller average SS values are usually obtained when LLMs demonstrate strong performance, larger SS values are observed even with fewer answer choices (i.e. without options E and F) when LLMs exhibit weaker performance (e.g., 3.57 vs. 3.74 for MPT-7B and 3.48 vs. 3.55 for Falcon-40B). It is also noted that Llama-2-70B, DeepSeek-67B and Yi-6B achieve the same average SS value of 1.89 when options E and F are excluded, making them not differentiable in terms of prediction uncertainty.

Table 12: The ratio of test instances for which the predicted answer is option E ("*I don't know*") or option F ("*None of the above*"). Note that neither of them corresponds to the ground truth answer.

| LLMs | E Rate (%) | | | | | | F Rate (%) | | | | | |
| --- | --- | --- | --- | --- | --- | --- | --- | --- | --- | --- | --- | --- |
| | QA | RC | CI | DRS | DS | Avg. | QA | RC | CI | DRS | DS | Avg. |
| Yi-34B | 1.79 | 0.41 | 0.00 | 1.99 | 0.00 | 0.84 | 0.37 | 0.08 | 0.07 | 0.07 | 0.00 | 0.12 |
| Qwen-72B | 0.31 | 0.47 | 0.39 | 1.53 | 0.00 | 0.54 | 0.32 | 0.62 | 1.19 | 3.22 | 0.02 | 1.07 |
| Qwen-14B | 0.21 | 0.83 | 0.08 | 0.19 | 0.00 | 0.26 | 0.43 | 0.70 | 0.17 | 0.19 | 0.27 | 0.35 |
| Llama-2-70B | 0.11 | 0.03 | 0.00 | 0.47 | 0.72 | 0.27 | 0.05 | 0.07 | 0.00 | 0.17 | 0.00 | 0.06 |
| DeepSeek-67B | 3.46 | 2.95 | 0.75 | 0.81 | 0.00 | 1.60 | 0.04 | 0.00 | 0.00 | 0.00 | 0.02 | 0.01 |
| Yi-6B | 5.88 | 1.01 | 0.00 | 5.71 | 0.00 | 2.52 | 0.05 | 0.15 | 0.00 | 0.03 | 0.00 | 0.05 |
| Gemma-7B | 0.61 | 0.03 | 0.00 | 0.03 | 0.04 | 0.14 | 0.00 | 0.00 | 0.00 | 0.00 | 0.00 | 0.00 |
| Mistral-7B | 0.18 | 0.00 | 0.00 | 0.01 | 0.00 | 0.04 | 0.00 | 0.00 | 0.00 | 0.00 | 0.00 | 0.00 |
| Llama-2-13B | 0.99 | 0.05 | 0.00 | 0.00 | 0.00 | 0.21 | 0.16 | 0.00 | 0.00 | 0.00 | 0.00 | 0.03 |
| Qwen-7B | 1.25 | 0.51 | 0.00 | 0.32 | 0.00 | 0.42 | 1.01 | 0.74 | 0.00 | 0.04 | 0.00 | 0.36 |
| InternLM-7B | 1.05 | 0.00 | 0.03 | 0.07 | 2.71 | 0.77 | 0.00 | 0.00 | 0.00 | 0.00 | 1.61 | 0.32 |
| Llama-2-7B | 0.39 | 0.00 | 0.00 | 0.00 | 1.85 | 0.45 | 0.01 | 0.00 | 0.00 | 0.00 | 0.12 | 0.03 |
| DeepSeek-7B | 1.05 | 1.67 | 0.00 | 0.00 | 0.39 | 0.62 | 0.58 | 0.00 | 0.00 | 0.00 | 0.00 | 0.12 |
| Qwen-1.8B | 1.65 | 0.51 | 0.68 | 0.00 | 1.01 | 0.77 | 0.00 | 0.00 | 0.00 | 0.00 | 0.52 | 0.10 |
| Falcon-40B | 0.15 | 0.01 | 0.00 | 0.00 | 3.72 | 0.78 | 0.00 | 0.05 | 0.05 | 0.00 | 0.95 | 0.21 |
| MPT-7B | 0.05 | 0.00 | 0.00 | 0.00 | 0.04 | 0.02 | 0.00 | 0.00 | 0.00 | 0.00 | 0.00 | 0.00 |
| Falcon-7B | 0.16 | 0.05 | 0.00 | 0.01 | 0.03 | 0.05 | 0.00 | 0.01 | 0.00 | 0.00 | 0.10 | 0.02 |

Table 13: The average prediction set size (SS) when the option E ("*I don't know*") and option F ("*None of the above*") are included in or excluded from the answer choices. Note that neither of them corresponds to the ground truth answer.

| LLMs | With Options E and F | | | | | | Without Options E and F | | | | | |
| --- | --- | --- | --- | --- | --- | --- | --- | --- | --- | --- | --- | --- |
| | QA | RC | CI | DRS | DS | Avg. | QA | RC | CI | DRS | DS | Avg. |
| Yi-34B | 2.60 | 1.71 | 1.90 | 1.77 | 1.69 | 1.93 | 2.06 | 1.42 | 1.52 | 1.49 | 1.58 | 1.61 |
| Qwen-72B | 2.45 | 1.90 | 1.80 | 2.09 | 2.06 | 2.06 | 2.02 | 1.52 | 1.53 | 1.59 | 1.78 | 1.69 |
| Qwen-14B | 2.80 | 1.74 | 2.02 | 1.94 | 2.37 | 2.17 | 2.39 | 1.41 | 1.54 | 1.67 | 2.04 | 1.81 |
| Llama-2-70B | 2.62 | 1.78 | 1.82 | 2.34 | 2.25 | 2.16 | 2.30 | 1.51 | 1.69 | 1.88 | 2.07 | 1.89 |
| DeepSeek-67B | 2.65 | 1.54 | 2.43 | 1.89 | 2.25 | 2.15 | 2.21 | 1.40 | 2.11 | 1.68 | 2.05 | 1.89 |
| Yi-6B | 3.20 | 1.92 | 1.88 | 2.85 | 1.96 | 2.36 | 2.41 | 1.54 | 1.77 | 1.99 | 1.76 | 1.89 |
| Gemma-7B | 2.72 | 1.88 | 2.04 | 2.14 | 3.11 | 2.38 | 2.42 | 1.69 | 1.97 | 2.05 | 2.96 | 2.22 |
| Mistral-7B | 2.80 | 1.75 | 2.48 | 2.71 | 2.40 | 2.43 | 2.48 | 1.67 | 2.48 | 2.53 | 2.28 | 2.29 |
| Llama-2-13B | 3.06 | 2.24 | 2.72 | 2.55 | 2.24 | 2.56 | 2.92 | 2.00 | 2.69 | 2.50 | 2.18 | 2.46 |
| Qwen-7B | 3.26 | 2.15 | 2.28 | 2.51 | 2.92 | 2.63 | 2.78 | 1.72 | 2.15 | 2.16 | 2.84 | 2.33 |
| InternLM-7B | 3.49 | 2.19 | 3.28 | 3.63 | 4.47 | 3.41 | 3.34 | 2.08 | 3.47 | 3.13 | 3.69 | 3.14 |
| Llama-2-7B | 3.20 | 2.39 | 3.27 | 3.26 | 3.30 | 3.09 | 3.45 | 2.36 | 3.57 | 3.57 | 2.93 | 3.18 |
| DeepSeek-7B | 3.34 | 2.77 | 3.06 | 3.40 | 3.08 | 3.13 | 3.41 | 2.42 | 3.45 | 3.61 | 3.23 | 3.22 |
| Qwen-1.8B | 3.20 | 2.58 | 3.49 | 3.45 | 4.18 | 3.38 | 3.43 | 2.36 | 3.59 | 3.46 | 3.71 | 3.31 |
| Falcon-40B | 3.25 | 3.12 | 3.54 | 3.59 | 3.89 | 3.48 | 3.46 | 3.20 | 3.70 | 3.72 | 3.66 | 3.55 |
| MPT-7B | 3.53 | 3.46 | 3.60 | 3.59 | 3.66 | 3.57 | 3.73 | 3.68 | 3.75 | 3.75 | 3.78 | 3.74 |
| Falcon-7B | 3.90 | 3.60 | 3.66 | 3.64 | 3.92 | 3.75 | 3.76 | 3.76 | 3.76 | 3.76 | 3.77 | 3.76 |

Consequently, the inclusion of these additional options does not significantly impact the accuracy of the evaluation process. This indicates that our approach to incorporating options E and F effectively increases the difficulty of the tasks without compromising the reliability of the assessment. With more options in the answer choices, we can also quantify uncertainty more accurately.

## C.10 Comparison of prompting strategies

In § 5, we have introduced three prompting strategies and our previous analyses are based on the average results obtained from these prompting strategies, aiming to reduce the influence of LLMs' sensitivities to prompts. In this subsection, we delve deeper into the comparative performance of these prompting strategies. Specifically, we conduct experiments on the DS task and report the results of Yi-34B, Qwen-72B, Llama-2-70B, and DeepSeek-67B in Table 14. It can be observed that while the

Table 14: Comparison of different prompting strategies on the DS task. The reported values of the SS metric are obtained using LAC as the conformal score function.

| LLMs | Prompting Strategy | Acc (%) | SS |
|---|---|---|---|
| Yi-34B | Base Prompt | **73.19** | **1.40** |
| | Shared Instruction Prompt | 69.87 | 1.46 |
| | Task-specific Instruction Prompt | 71.35 | 1.42 |
| Qwen-72B | Base Prompt | 58.30 | 1.70 |
| | Shared Instruction Prompt | 61.08 | 1.67 |
| | Task-specific Instruction Prompt | **62.51** | **1.61** |
| Llama-2-70B | Base Prompt | 56.66 | **2.08** |
| | Shared Instruction Prompt | 56.18 | 2.21 |
| | Task-specific Instruction Prompt | **59.38** | 2.18 |
| DeepSeek-67B | Base Prompt | 55.60 | 2.10 |
| | Shared Instruction Prompt | **56.66** | **2.03** |
| | Task-specific Instruction Prompt | 56.34 | 2.18 |

performance of each LLM varies with different prompting strategies, the discrepancies are generally marginal. Of greater significance is the observation that different LLMs demonstrate preferences for different prompting strategies. For example, the base prompting strategy yields the highest accuracy for Yi-34B. Conversely, Qwen-72B performs optimally with the task-specific instruction prompting strategy, while DeepSeek-67B exhibits superior accuracy with the shared instruction prompting strategy. Similar patterns can be observed from the results of the SS metric. These observations highlight the importance of considering multiple prompting strategies when benchmarking LLMs. By averaging the results from various strategies, we can mitigate the impact of prompt sensitivities and ensure a more equitable comparison of LLM performance.

## C.11 Ablation study

Inspired by in-context learning [66], our previous analyses have incorporated demonstrations for each task. Here, we aim to compare the performance of LLMs when demonstrations are included or excluded from the prompt input. The results of Yi-34B and Llama-2-70B are reported in Table 15. We observe that the presence of demonstrations leads to improved performance for both models, as evidenced by increased accuracy. Having demonstrations also leads to lower uncer-

Table 15: Average results across five tasks of Yi-34B and Llama-2-70B. $*$ indicates the results obtained without using demonstrations in the prompt input.

| LLMs | CR (%) | Acc (%) $\uparrow$ | SS $\downarrow$ |
|---|---|---|---|
| Yi-34B | 94.22 | 81.46 | 1.93 |
| Yi-34B$*$ | 93.31 | 80.10 | 1.88 |
| Llama-2-70B | 93.11 | 72.24 | 2.16 |
| Llama-2-70B$*$ | 91.55 | 63.55 | 2.72 |

tainty for Llama-2-70B but higher uncertainty for Yi-34B. Overall, these results substantiate the effectiveness of incorporating demonstrations into the prompt to stimulate the capabilities of LLMs.

## C.12 Case study

Table 16 presents an example of the prediction sets produced by the Yi-34B model on the QA task. It is noteworthy that the correct answer is always encompassed in the prediction sets, irrespective of the employed prompting strategies (including base prompt, shared instruction prompt, and task-specific instruction prompt) and conformal score functions (including LAC and APS). In this specific example, we further observe that the LAC score function consistently produces prediction sets with smaller sizes compared to APS, with the prediction sets only containing the correct answer. However, the prediction sets generated by APS can exhibit variations dependent on the prompting strategies. In this particular example, we observe that different prompting strategies result in different prediction sets when APS is utilized as the conformal score function. This case study reveals that varying conformal score functions and prompting strategies can yield different measurements of uncertainty, even though the true answer is encompassed in the prediction sets. In practice, the mean results

Table 16: An example of the prediction sets produced by the Yi-34B model on the QA task. We include results derived from both the LAC and APS score functions and the three prompting strategies. All generated prediction sets encompass the true answer.

---

**Question:** Which of the following is thought to be implicated in the development of peripheral muscle fatigue during multiple sprint activities?
**Choices:**
A. *An accumulation of inorganic phosphate.*
B. *Development of hyperosmolality in the muscles.*
C. *An excess of antioxidants.*
D. *A lack of potassium.*
E. *I don't know*
F. *None of the above*
**Correct Answer:** A
**Predicted Answer based on LAC:**
  Base Prompt: {A}
  Shared Instruction Prompt: {A}
  Task-specific Instruction Prompt: {A}
**Predicted Answer based on APS:**
  Base Prompt: {A, B, D}
  Shared Instruction Prompt: {A, F}
  Task-specific Instruction Prompt: {A, E}

---

derived from these conformal score functions and prompting strategies can be used to achieve a more precise uncertainty quantification.

## D  Prompt templates

We provide the prompt templates employed by the three prompting strategies in Table 17, Table 18, and Table 19, respectively. For the QA task, there is no background information for each question. For the RC and CI tasks, each question has an associated contextual description, as indicated by the keyword "Context" in the prompt templates. For the DRS task and the DS task, we use "Dialogue" and "Document" rather than "Context" as the keywords to incorporate the dialogue history and document content into the prompt. When evaluating instruction-finetuned LLMs, we treat the entire prompt input as the message from users and then employ the "*apply_chat_template*" function to transform the prompt input into a chat format.

## E  Limitations

Despite the multiple advantages of conformal prediction over other uncertainty quantification methods, it demonstrates three key limitations when employed to assess the uncertainty of LLMs. Firstly, the application of conformal prediction necessitates access to model output logits, which precludes the possibility of benchmarking LLMs such as ChatGPT that are only accessible via their APIs. Secondly, the adoption of conformal prediction poses challenges in evaluating the generative capabilities of LLMs. In our analyses, all tasks are transformed into multiple-choice questions, thereby primarily assessing the language understanding abilities of LLMs rather than their generative potential. Thirdly, the prediction sets generated by conformal prediction could be significantly influenced by the conformal score function utilized. Thus, for a specific LLM, varying conformal score functions may yield disparate estimations of uncertainty. It is worth mentioning that while we have extended our benchmarking to closed-source LLMs and free-form text generation, the extension can only provide an approximation.

Nevertheless, the limitations of other uncertainty quantification methods prevent us from applying them in the context of LLMs. Per our knowledge, conformal prediction is currently the most feasible technique for *robust* uncertainty quantification of LLMs. Furthermore, some recent studies have tried to incorporate conformal prediction into the language generation process [55, 53, 16]. We posit that conformal prediction will eventually evolve into an appropriate method for quantifying the uncertainty of language generation in the future.

Last but not least, it is important to note that the scope of this study is limited to evaluating the capabilities of LLMs in the context of language processing exclusively. The current trend in the field is towards the development of multi-modal foundation models [44, 41, 47], which have the capacity to process multiple modalities rather than just language. Therefore, it would be a significant extension of this research to investigate the uncertainty associated with these foundation models when they are applied to non-language modalities, which constitutes an important avenue for future research.

## F    Societal Impacts

While benchmarking LLMs via uncertainty quantification can lead to more accurate assessments of their performance, it is crucial to consider and address any potential negative societal impacts. We list several possible concerns below:

- **Misuse of Technology:** Enhanced performance and reliability of LLMs could lead to their misuse in generating misleading or harmful content, such as deepfakes, disinformation, or automated trolling. Improved uncertainty quantification might make these models more convincing and harder to detect.

- **Bias and Fairness:** Even with improved uncertainty quantification, LLMs can still perpetuate and amplify existing biases present in the training data. This can lead to unfair treatment of certain groups or individuals, reinforcing stereotypes and discrimination.

- **Job Displacement:** As LLMs become more capable, there is a potential for job displacement in fields that rely heavily on language processing, such as customer service, translation, and content creation. This could lead to economic and social challenges for affected workers.

Table 17: Prompt template for the base prompting strategy. Note that for the QA task, there is no background information pertaining to questions. For the RC and CI tasks, we adopt the keyword "Context" to include the contextual information associated with each question. For the DRS and DS tasks, we employ the keywords "Dialogue" and "Document" to incorporate the pertaining information, respectively.

```
{Demonstrations in the same format as the question shown below except
 that the true answers are provided for questions in the demonstrations}
Context/Dialogue/Document:  {The context or dialogue history or
document corresponding to the following question}
Question:  {Question}
Choices:
A. {Content of option A}
B. {Content of option B}
C. {Content of option C}
D. {Content of option D}
E. I don't know
F. None of the above
Answer:
```

Table 18: Prompt template for the shared instruction prompting strategy. In this strategy, we add a shared general task description at the beginning of the prompt. Note that for the QA task, there is no background information pertaining to questions. For the RC and CI tasks, we adopt the keyword "Context" to include the contextual information associated with each question. For the DRS and DS tasks, we employ the keywords "Dialogue" and "Document" to incorporate the pertaining information, respectively.

```
Below are some examples of multiple-choice questions with six potential
answers.  For each question, only one option is correct.

{Demonstrations in the same format as the question shown below except
 that the true answers are provided for questions in the demonstrations}

Now make your best effort and select the correct answer for the
following question.  You only need to output the option.

Context/Dialogue/Document:  {The context or dialogue history or
document corresponding to the following question}
Question:  {Question}
Choices:
A. {Content of option A}
B. {Content of option B}
C. {Content of option C}
D. {Content of option D}
E. I don't know
F. None of the above
Answer:
```

Table 19: Prompt template for the task-specific instruction prompting strategy. In this strategy, we add a task-specific description at the beginning of the prompt. Note that for the QA task, there is no background information pertaining to questions. For the RC and CI tasks, we adopt the keyword "Context" to include the contextual information associated with each question. For the DRS and DS tasks, we employ the keywords "Dialogue" and "Document" to incorporate the pertaining information, respectively.

```
(for the QA task) Below are some examples of multiple-choice questions
about question answering.  Each question should be answered based on
your world knowledge and problem solving ability./
(for the RC task) Below are some examples of multiple-choice questions
about reading comprehension.  Each question should be answered based on
the given context and commonsense reasoning when necessary./
(for the CI task) Below are some examples of multiple-choice questions
about commonsense natural language inference.  For each question,
there is a given context and the answer is the option that most likely
follows the context./
(for the DRS task) Below are some examples of multiple-choice questions
about dialogue response selection.  For each question, the answer is
the option that represents the most suitable response for the given
dialogue history, without hallucination and non-factual information./
(for the DS task) Below are some examples of multiple-choice questions
about document summarization.  For each question, the answer is
the option that accurately summarizes the given document without
hallucination and non-factual information.

{Demonstrations in the same format as the question shown below except
that the true answers are provided for questions in the demonstrations}

Now make your best effort and select the correct answer for the
following question.  You only need to output the option.

Context/Dialogue/Document:  {The context or dialogue history or
document corresponding to the following question}
Question:  {Question}
Choices:
A. {Content of option A}
B. {Content of option B}
C. {Content of option C}
D. {Content of option D}
E. I don't know
F. None of the above
Answer:
```

