# OpenReview forum: "Benchmarking LLMs via Uncertainty Quantification"
_NeurIPS.cc/2024/Datasets_and_Benchmarks_Track — NeurIPS 2024 Track Datasets and Benchmarks Poster_

### Official Review · Reviewer_MsFo · 2024-07-13
**It's interesting, but it still requires substantial effort before being published.**

**Rating:** 6
**Confidence:** 4
**Clarity:** The paper is easy to understand.

**Review:**

Overall, this study is straightforward to understand. The uncertainty quantification of LLMs is also an important topic in the AI community, and the paper has several interesting findings. However, there are several major concerns regarding this study in terms of its suitability to this track, its significance, and its soundness.

While this paper claims to be an LLM benchmark via quantification, I find its motivation and aim relatively strange. From my understanding, although this paper emphasizes several times that the conformal prediction-based uncertainty measurement is a distribution-free and statistically rigorous approach, it is still an uncertainty estimation method.
There is still a (perhaps large) gap between the computed uncertainty scores and the real "uncertainty" of the model. So, it is unclear whether the obtained results really have something to do with the models' capabilities. I would say benchmarking different uncertainty estimation methods (instead of LLMs) for their effectiveness in various downstream tasks is a more meaningful topic. What's more, even if we can obtain completely accurate uncertainty scores of LLMs, we still cannot use such information to "judge" the quality of LLMs. An LLM with higher uncertainty does not mean that their performance is bad, and related stakeholders should try to lower their uncertainty. The LLMs' uncertainty metric differs from performance or safety/security-related properties. Regarding this paper, I think it is more like an exploratory empirical study rather than a benchmark paper with a new dataset. However, several related studies have already explored the uncertainty quantification of LLMs, and thus, this paper is not so novel.

Another significant limitation is that this paper focuses totally on multiple-choice classification tasks. However, LLMs are well-known for their autoregressive generation nature, and free-form style response is a very important characteristic. Technically speaking, how to define and measure uncertainty scores using conformal prediction for free-form content is much more challenging and meaningful.

Regarding the soundness of this paper, there are several concerns. This paper mentioned multiple times that conformal prediction is capable of performing distribution-free, model-agnostic, and statistically rigorous estimation. While it is trivial to see the estimation as model-agnostic, the authors do not provide definitions of distribution-free and statistically rigorous. For the former, I saw there is a held-out calibration dataset for equation (1). For the latter, I do not see any mathematical proof regarding the theoretical guarantee of the proposed metric. It is very hard to give such a guarantee, given that the analysis is still based on softmax and is inevitably empirical.

As such, I think this paper should be extensively revised before publication.

**Strengths:**

+ An interesting way to estimate uncertainty scores of LLMs
+ An extensive study involving nine LLMs and five diverse NLP tasks.

**Additional Feedback:**

This benchmark includes a document summarization task. From my understanding, this task itself is not classification-oriented. Could you explain how you transform it into a multiple-choice question setting?

**Correctness:**

The benchmark construction is generally sound. However, the way the authors describe their uncertainty measurement methods should be further improved for soundness.

**Documentation:**

This paper is well-documented.

**Ethics:**

No, there is no such concern for this study.

**Limitations:**

- The motivation of the paper should be further justified. The current version is somehow unclear for the benchmark track.

**Opportunities For Improvement:**

- It might be better to add more in-depth analysis regarding the proposed metrics to improve the significance of the paper.
- It might be better to study free-form generation instead of classification tasks.
- The authors should provide clearer definitions of several terminologies in the paper.

**Relation To Prior Work:**

Possibly yes.

**Summary And Contributions:**

This paper presents a conformal prediction-based uncertainty quantification benchmark for LLMs. Given two conformal score functions, namely, LAC(Least Ambiguous set-valued Classifiers) and APS (Adaptive Prediction Sets), this benchmark systematically evaluated nine LLMs on five representative NLP tasks. Evaluation results reveal several interesting findings, including the relationship between uncertainty scores and models' performance and the influence of instruction finetuning. The major contribution of this paper is the large-scale study on conformal prediction-based uncertainty quantification and related result analysis.

---

> ### Author Rebuttal · Authors · 2024-08-23
>
> Dear Reviewer,
>
> Thank you for your insightful comments and appreciation of our work. We apologize for the slight delay in our response, as it took us a considerable amount of time to complete the experiments. Below, we address your concerns and suggestions.
>
> **Q1:** *Add more in-depth analysis regarding the proposed metrics.*
>
> **A1:** Thanks for this suggestion. We agree that capturing the "real" uncertainty of models is indeed a challenging task. This is why a variety of uncertainty quantification methods have been developed within the machine learning community, with the aim of enhancing the accuracy of uncertainty quantification.
>
> In our study, we have chosen to adopt conformal prediction as our uncertainty quantification method. Conformal prediction holds several advantages over other methods, including its ease of implementation, distribution-free and model-agnostic nature, and its statistically rigorous estimation of uncertainty (with model accuracy taken into account). This means that **conformal prediction can offer more reliable uncertainty quantification**.
>
> Additionally, conformal prediction generates a prediction subset and uses the size of this subset to measure uncertainty. This approach is **not only intuitive but also highly interpretable, which further enhances the reliability of the uncertainty measured by conformal prediction**.
>
> In our paper, **we have compared conformal prediction with entropy in Section 6.6 and Appendix C.7**. The results indicate that entropy is highly sensitive to the temperature of the softmax function and unlike conformal prediction, entropy cannot provide any guarantee on the coverage rate. This is because entropy does not consider model accuracy when measuring uncertainty.
>
> However, to further validate the superiority of conformal prediction, we conduct an additional experiment comparing three uncertainty quantification methods: **conformal prediction (CP)**, **entropy**, and **maximal predicted probability ($P_{max}$)**. We use the **expected calibration error (ECE)** metric [1] to evaluate the reliability of these methods. The results, which correspond to InternLM-7B, are presented below:
> | Method   | QA    | RC   | CI   | DRS  | DS   | Avg. |
> |----------|:-------:|:------:|:------:|:------:|:------:|:------:|
> | CP       | **15.83** | 1.33 | **3.16** | **12.11** | **9.30** | **8.35** |
> | Entropy  | **15.83** | **1.32** | 3.45 | 12.40 | **9.30** | 8.46 |
> | $P_{max}$    | **15.83** | 1.41 | 3.75 | 12.45 | 9.62 | 8.61 |
>
> The results indicate that, overall, conformal prediction provides more reliable uncertainty estimation compared to the other methods.
>
> **Q2:** *An LLM with higher uncertainty does not mean that their performance is bad, and related stakeholders should try to lower their uncertainty.*
>
> **A2:** We appreciate this comment and agree with the point that an LLM with high uncertainty may exhibit high accuracy. In fact, this is our motivation of this work as **we intend to examine the performance of different LLMs more comprehensively by considering both model accuracy and model uncertainty**. Conformal prediction is advantageous in this regard, as it accounts for both model uncertainty and model accuracy simultaneously, which is why we have adopted this approach. **We concur that relying solely on uncertainty is insufficient for judging the quality of LLMs**.
>
> However, we respectfully disagree with the notion that we should not attempt to reduce the uncertainty of models. Gaining an understanding of the uncertainty associated with an LLM's outputs can yield valuable insights. For example, it can help identify areas where the model may benefit from additional training or fine-tuning. Furthermore, it can be advantageous in applications involving decision-making, as it can inform users about the confidence level of the model's predictions. When comparing two models with equal accuracy but different uncertainties, the one with lower uncertainty should be preferred.
>
> **By focusing on both accuracy and uncertainty, we aim to provide a more comprehensive evaluation of LLMs and contribute to the development of models that are not only accurate but also reliable in their predictions**.
>
> **Q3:** *It might be better to study free-form generation instead of classification tasks.*
>
> **A3:** We believe that the study of classification tasks carries significant importance, considering the fact that most existing benchmarking datasets are structured in the form of multiple-choice questions. Our method of uncertainty quantification can be smoothly incorporated into these datasets.
>
> At the same time, we acknowledge the value of studying free-form generation. However, the application of conformal prediction to text generation presents a complex challenge, given the vast array of potential responses. It's not feasible to compute the probability for each possible response and then use conformal prediction to select a subset. However, many potential responses have a low probability of being generated, which allows us to reduce the selection space by sampling multiple generations.
>
> Specifically, **we adopt the TriviaQA dataset [2] (sampling 10K instances) for freeform text generation**. We generate 20 answers for each question. We then utilize the perplexity of each generation as the conformal score function and use exact match to verify the accuracy of the generated answer. The results are displayed in the table below.
> | LLMs        | Accuracy (Acc, %) (↑) |  Prediction Uncertainty (SS) (↓) | Coverage Rate (CR, %)    |
> |-------------|:-------:|:------:|:-------:|
> | Qwen-72B    | 76.45 | 2.63 | 88.92 |
> | Llama-2-13B | 71.83 | 2.40 | 83.89 |
> | Qwen-14B    | 66.57 | 3.83 | 82.79 |
> | Llama-2-7B  | 64.91 | 3.06 | 78.83 |
> | Qwen-7B     | 59.44 | 5.02 | 77.89 |
> | DeepSeek-7B | 57.52 | 6.12 | 78.41 |
> | Falcon-7B   | 55.74 | 6.27 | 76.51 |

---

> > ### Author Rebuttal · Authors · 2024-08-23
> >
> > **A3 (Cont.):** It can be seen that the prediction set size (SS) varies among models, which could provide some insight into the uncertainty of these models. However, we must note that in this sampling setting, the coverage rate cannot be guaranteed because there might not be a correct answer within the 20 sampled responses. In other words, even if the prediction set size is 20, indicating high model uncertainty, the coverage rate for that instance could still be zero if there are no correct answers present.
> >
> > We aim to develop more effective algorithms for applying conformal prediction to freeform text generation. However, this is a complex task and its challenges are widely acknowledged in the research community.
> >
> > **Q4:** *The authors should provide definitions of distribution-free and statistically rigorous. For the former, I saw there is a held-out calibration dataset for equation (1). For the latter, I do not see any mathematical proof regarding the theoretical guarantee. It is very hard to give such a guarantee, given that the analysis is still based on softmax.*
> >
> > **A4:** Thanks for this suggestion. We agree that it would be beneficial to provide clear definitions of distribution-free and statistically rigorous properties of conformal prediction in our paper. **We have referred to relevant literature, such as [3], for a detailed explanation of these properties**. However, we will include the following explanations in the final version of our paper to provide a better understanding:
> >
> > - **Distribution-free:** Conformal prediction is not dependent on any specific distributional assumptions about the data. This means it can effectively handle various types of data distributions, whether it is normal, skewed, or heavy-tailed. The distribution-free property is a result of conformal prediction's reliance on exchangeability and the use of nonconformity measures. Nonconformity measures assess how different a data point is from others in the dataset without making assumptions about the data distribution. By comparing the nonconformity measures of a new data point to those of the calibration set, conformal prediction constructs prediction sets with associated confidence levels or coverage guarantees, irrespective of the underlying data distribution.
> >
> > - **Statistically rigorous:** Conformal prediction offers a theoretical guarantee on the coverage rate of prediction sets. This implies that, for a given confidence level (e.g., 90%), the true label of a new data point will be included in the prediction set with a probability of at least 90%. This guarantee is valid for any data distribution and any model, making conformal prediction statistically rigorous. **A formal proof of this property can be found in Appendix D of [3]**.
> >
> > Due to the statistically rigorous property, the estimation remains rigorous even when based on softmax output. **Conformal prediction effectively converts the heuristic notion of uncertainty derived from softmax output into a rigorous notion of uncertainty. This is one of the biggest advantages of conformal prediction**.
> >
> > **Q5:** *Could you explain how you transform the summarization task into a multiple-choice question setting?*
> >
> > **A5:** Thanks for this question. **The details are provided in Section 4 of our submission**. We construct this dataset on top of HaluSum [4], which is originally in the form of a multiple-choice question setting.
> >
> > Hope that we have addressed all your concerns. Should you have any further questions, please kindly let us know.
> >
> > [1] GUO, Chuan, et al. On calibration of modern neural networks. ICML 2017.
> >
> > [2] https://nlp.cs.washington.edu/triviaqa/
> >
> > [3] Anastasios N. Angelopoulos and Stephen Bates. **A gentle introduction to conformal prediction and distribution-free uncertainty quantification**. arXiv preprint arXiv:2107.07511, 2021.
> >
> > [4] LI, Junyi, et al. HaluEval: A Large-Scale Hallucination Evaluation Benchmark for Large Language Models. EMNLP 2023.

---

> > > ### Comment · Reviewer_MsFo · 2024-08-24
> > >
> > > I appreciate the authors' efforts and the updates made to the manuscript. However, I would suggest being cautious when using terms such as "statistically rigorous" and "distribution-free." In the context of conformal uncertainty prediction, the approach inherently relies on calibrating measurements using an additional dataset. As a result, the predictions cannot be formally guaranteed to be "correct," as they are empirically derived from an auxiliary "training" dataset. Such techniques may struggle with out-of-distribution scenarios and can significantly diverge from related "statistically rigorous" definitions within the AI community, particularly concerning AI security (robustness) and fairness.
> > >
> > > However, since this paper focuses on benchmarking existing methods rather than proposing a new one, this concern is relatively limited and not a major issue.

---

> > > > ### Author Rebuttal · Authors · 2024-08-24
> > > >
> > > > Thanks for this suggestion. We will consider replacing or removing the two terms in our final version.
> > > >
> > > > Indeed, in conformal prediction, the calibration data and test data need to be exchangeable. When the test data follow a different distribution from the calibration data, the estimation accuracy may be compromised. To address this issue, we can consider the following strategies:
> > > >
> > > > - **Resampling and recalculating:** We can resample the calibration data and recalculate the conformal threshold to better adapt to the test data distribution. As demonstrated in **Appendix C.6** of our submission, a small calibration set is sufficient to yield accurate uncertainty estimation, which makes this resampling and recalculating procedure feasible.
> > > >
> > > > - **Distribution shift extensions:** We can leverage the existing research such as [5] that extends standard conformal prediction to handle distribution shifts. By incorporating these extensions, our benchmarking can better adapt to cases where the calibration and test data follow different distributions.
> > > >
> > > > We will incorporate these considerations as our future work in the final version.
> > > >
> > > > [5] GIBBS, Isaac; CANDES, Emmanuel. Adaptive conformal inference under distribution shift. NeurIPS 2021.

---

> > > ### Comment · Reviewer_MsFo · 2024-08-24
> > >
> > > I appreciate the new experiments conducted on TriviaQA and other explanations in the rebuttal. I will raise my score to 5 since they address some of my concerns.

---

> > > > ### Author Rebuttal · Authors · 2024-08-24
> > > >
> > > > Much appreciated!!!

---

> ### Comment · Reviewer_MsFo · 2024-08-24
>
> > However, we respectfully disagree with the notion that we should not attempt to reduce the uncertainty of models.
>
> There might be some misunderstandings regarding my previous comments. My point is not that "we should avoid reducing model uncertainty," but rather, I am curious about the objectives behind your uncertainty benchmark. Specifically, what do you aim to achieve with your uncertainty measurements? If a model is uncertain about a particular question, we would naturally expect it to output a higher uncertainty score compared to those questions that fall in the model's knowledge scope. Therefore, the real focus of uncertainty measurement should be on its accurate estimation (of the unknown, ground truth uncertainty). This question is closely related to the underlying motivation of your study and is perhaps one of my major concerns.

---

> > ### Author Rebuttal · Authors · 2024-08-24
> >
> > Dear Reviewer,
> >
> > Thank you for your prompt reply. Our primary goal is to provide a more comprehensive performance evaluation of LLMs by considering both model accuracy and model uncertainty. This is in contrast to most existing benchmark methods that focus solely on model accuracy.
> >
> > To achieve this goal, we aim to assess whether LLMs with higher accuracy exhibit lower uncertainty. Ideally, when a model's prediction is correct, its uncertainty should be low, and conversely, when the prediction is incorrect, its uncertainty should be high. By benchmarking LLMs, we have observed that some models with higher accuracy demonstrate lower certainty, which is not ideal.
> >
> > We fully agree with your point regarding the importance of accurate uncertainty estimation. That is why we have chosen conformal prediction as our uncertainty quantification method. Conformal prediction provides more reliable uncertainty estimation compared to many other methods such as entropy. Its coverage guarantee requirement enables the method to consider both model accuracy and model uncertainty. This property is particularly useful when a model has high confidence in its prediction, but the prediction is incorrect. Conformal prediction can identify such cases and produce a large prediction set (implying high uncertainty) due to the requirement of covering the true answer.
> >
> > To further support our choice, we have conducted an additional experiment to show that **a smaller prediction set (indicating less uncertainty) indeed often correlates with higher accuracy**. We adopt InternLM-7B as the LLM and conduct experiments on the QA task. The results are shown in the table below:
> > | Prediction Set Size | LAC   | APS   | Avg.  |
> > |---------------------|:-------:|:-------:|:-------:|
> > | 1                   | 80.39 | 92.69 | 86.54 |
> > | 2                   | 59.77 | 82.21 | 70.99 |
> > | 3                   | 40.55 | 63.70 | 52.12 |
> > | 4                   | 40.07 | 41.71 | 40.89 |
> > | 5                   | 31.50 | 34.92 | 33.21 |
> > | 6                   | 13.43 | None  | 13.42 |

---

> > > ### Comment · Reviewer_MsFo · 2024-08-28
> > >
> > > Thanks for your update. While some of my concerns still exist, I carefully reconsider the main content of the paper and think that, at this time, this might be a meaningful attempt, and we should encourage related studies along the way. As such, I will adjust my final score to 6.

---

> > > > ### Author Rebuttal · Authors · 2024-08-29
> > > >
> > > > Thanks a lot!!!

---

### Official Review · Reviewer_zVYp · 2024-07-23
**Useful benchmark with some limitations**

**Rating:** 7
**Confidence:** 3
**Clarity:** The paper and code are clear.

**Review:**

Overall this is a solid paper, and I agree with that authors main motivation that benchmarks are needed to assess the uncertainty of LLMs. The set of experiments is quite expansive with a lot of models and tasks considered. The use of conformal prediction makes sense given the large computational cost of Bayesian approaches, etc. The use of set size is nice since it provides an easily interpretable measure of the model's uncertainty.  The experiment on instruction fine-tuning is particularly interesting. This lines with prior work showing that fine-tuning models tend to more uncertain.

My biggest concern with this paper is that all the tasks are multi-choice classification problems. This is understandable as it simplifies the problem and makes the conformal prediction piece more straightforward. However, LLMs are not used this way in practice and its not clear how much training data of this form they saw.  It may be the case that the conclusions drawn here may not generalization to other use cases of LLMs. That said, its not clear how to easily move to the freeform generation case as the complexity would increase significantly.

**Strengths:**

A major strength of this work is the large number of models and datasets that were considered in the experiments.

**Additional Feedback:**

n/a

**Correctness:**

The experiments seem to be designed in a sound way. The use of conformal prediction is a natural choice and the overall results are straightforward to interpret.

**Documentation:**

The authors provide a link to the code which seems to have already have some traction.

**Ethics:**

There is no major ethical concerns for this work.

**Limitations:**

A major limitation of this work (as discussed above) is that it only considers classification tasks. It would help if the authors made some comment about this and what the results shown here might tell us about the general use case of LLMs with respect to uncertainty.

**Opportunities For Improvement:**

As discussed above, figuring out a way to move beyond classification seems critical going forward.

**Relation To Prior Work:**

There is little discussion of prior work, but the authors point out that rigorous benchmarks for UQ of LLMs does not yet exist. This is the main motivation of the paper.

**Summary And Contributions:**

This work is motivated by the fact that current LLM benchmarks do not account for the uncertainty that a model has over its predictions. The authors put forward a benchmark which uses conformal prediction to assess the uncertainty that a model has. They apply their approach to a number of current LLMs and NLP tasks. In this way they are able to characteristic the uncertainty of current models.

---

> ### Author Rebuttal · Authors · 2024-08-23
>
> Dear Reviewer,
>
> Thank you for your insightful comments and appreciation of our work. We apologize for the slight delay in our response, as it took us a considerable amount of time to complete the experiments. Below, we address your concerns and suggestions.
>
> **Q1:** *All the tasks are multi-choice classification problems. It's not clear how to easily move to the freeform generation case.*
>
> **A1:** Thanks for this comment. Indeed, applying conformal prediction to text generation is a complex task due to the extensive range of potential responses. It's not feasible to compute the probability for each possible response and then use conformal prediction to select a subset. However, many potential responses have a low probability of being generated, which allows us to reduce the selection space by sampling multiple generations.
>
> Specifically, **we adopt the TriviaQA dataset [1] (sampling 10K instances) for freeform text generation**. We generate 20 answers for each question. We then utilize the perplexity of each generation as the conformal score function and use exact match to verify the accuracy of the generated answer. The results are displayed in the table below:
> | LLMs        | Accuracy (Acc, %) (↑) |  Prediction Uncertainty (SS) (↓) | Coverage Rate (CR, %)    |
> |-------------|:-------:|:------:|:-------:|
> | Qwen-72B    | 76.45 | 2.63 | 88.92 |
> | Llama-2-13B | 71.83 | 2.40 | 83.89 |
> | Qwen-14B    | 66.57 | 3.83 | 82.79 |
> | Llama-2-7B  | 64.91 | 3.06 | 78.83 |
> | Qwen-7B     | 59.44 | 5.02 | 77.89 |
> | DeepSeek-7B | 57.52 | 6.12 | 78.41 |
> | Falcon-7B   | 55.74 | 6.27 | 76.51 |
>
> It can be seen that the prediction set size (SS) varies among models, which could provide some insight into the uncertainty of these models. However, we must note that in this sampling setting, the coverage rate cannot be guaranteed because there might not be a correct answer within the 20 sampled responses. In other words, even if the prediction set size is 20, indicating high model uncertainty, the coverage rate for that instance could still be zero if there are no correct answers present. Nevertheless, it is observed that when the model is stronger, the coverage guarantee requirement is more likely to be satisfied.
>
> We aim to develop more effective algorithms for applying conformal prediction to freeform text generation. However, this is a complex task and its challenges are widely acknowledged in the research community.
>
> Hope that we have addressed your concern. Should you have any further questions, please kindly let us know.
>
> [1] https://nlp.cs.washington.edu/triviaqa/

---

> > ### Author Rebuttal · Authors · 2024-08-28
> >
> > Dear Reviewer,
> >
> > As the discussion period is approaching the end, may we know if we have answered your questions? Thanks

---

### Official Review · Reviewer_XKVJ · 2024-07-25
**An effort to benchmark LLM uncertainty but results are only limited to conformal prediction and multiple choice questions.**

**Rating:** 7
**Confidence:** 5
**Correctness:** The claims made in this work seem cor…
**Clarity:** The paper is well written. The concep…

**Review:**

The paper is well written and easy to follow. The questions it raises and topics it covers are timely and of great importance to the Neurips audience. Below, I mention its strengths and limitations one by one.

**Strengths:**

1. The main strength of this paper is that it discusses a timely topic of LLM uncertainty. The main premise is that the accuracy alone may not be enough to inspect the performance of LLMs. So, the uncertainty of these models in generating their responses is of great importance if we want to get a deeper understanding of LLMs.

2. Though only conformal prediction is considered, authors provide a variety of results considering various tasks and LLM series. Results for the effects of mixture of experts and task unification are particularly interesting to me.

**Additional Feedback:**

Please refer to the opportunities for improvement section above.

**Documentation:**

Authors provide their benchmark implementation for reproducibility.

**Ethics:**

I do not suspect there are any ethical concerns with this submission.

**Limitations:**

Authors have adequately discussed the limitations of their work. I particularly agree with the authors that considering the uncertainty for multi modal models is an important future direction.

The potential negative societal impact of their work is not included.

**Opportunities For Improvement:**

Here are some points to improve:

1. As the authors also briefly mention, there are many uncertainty estimation/quantification methods for LLMs. Some of them include consistency-based methods, self-checking, the inspection of internal states/activations, token probability-based methods, i.e., entropy, semantic entropy, as well as the conformal prediction. In this landscape, authors only consider the conformal prediction approach in their benchmark. I understand the nature of this choice, as conformal prediction offers rigorous results. However, to me, a proper benchmarking of LLMs through uncertainty estimation/quantification should include representatives of other methods as well. I think this constitutes the biggest point for improvement in this work together with my next point.

2. As a result of their conformal prediction choice, authors restrict themselves with only multiple choice tasks. As the authors also mention, this essentially limits them to only benchmark LLMs' language understanding capabilities as opposed to their text generation capabilities. This is unlike the body of work that uses methods such as entropy and semantic entropy in their uncertainty analyses for general QA tasks.

3. While conformal prediction provides rigorous uncertainty quantification with certain coverage guarantees, the process of defining a conformal score is itself heuristic in nature, no less so than the computation of entropy or semantic entropy. Thus, I find the authors' characterization of other methods as mere heuristics and their dismissal from the current work rather hand-wavy. Further discussion on why these methods are not as good should be included along with demonstrations of their inefficiency.

4. The main hypothesis in the existing uncertainty estimation works for LLMs is that an LLM is likely to answer correctly if it is less uncertain. This is usually measured by the AUROC score such that higher AUROC scores imply that a model is better in distinguishing correct and incorrect answers. So, in a way, accuracy and uncertainty go together. In the current work, however, the set size metric does not give such intuition. This is also emphasized by the authors as one of their main results is that an LLM may achieve high accuracy, i.e., it can answer correctly, even if it is not certain. In order to accurately position the set size metric in the literature, I suggest including the AUROC metric as well.

5. In the implementation, each question has 6 possible answers where the last two choices (none of the above and I do not know choices) are not the ground truth labels. In such a scenario, from Table 1, we deduce that the average set size in the conformal prediction is larger than 3 on many tasks for many LLMs. This is particularly the case for the QA task. What to do with these conformal predictions? That is, when the model returns 3 or more choices as possible answers, are we really adequately chracterizing the uncertainty? Similarly, in the summarization task, the original dataset has only two choices for each question. Authors then append two random answers from other questions and add the "none of the above" and "I do not know" choices as well. In this case, we see again from Table 1, that the resulting set sizes are overwhelmingly larger than 2. In these cases, does this method effectively capture the uncertainty of the models?

**Relation To Prior Work:**

The prior work on uncertainty estimation/quantification in LLMs is not discussed in proper detail. I suggest authors to describe the existing literature in more detail in an effort to present the state-of-the-art more clearly as well as to justify the fact that they only use conformal prediction for uncertainty quantification purposes in their benchmark.

**Summary And Contributions:**

This paper aims at benchmarking the uncertainty of LLMs using a conformal prediction approach unlike the most of the literature that focuses solely on the accuracy of these models. Main conclusion of the paper is that accuracy alone may not be the best metric to judge the performance of LLMs, as often times these models may exhibit varying degrees of uncertainty in generating responses. Authors focus on a variety of tasks and LLM series in supporting their main findings.

---

> ### Author Rebuttal · Authors · 2024-08-20
>
> Dear Reviewer,
>
> Thank you for your insightful comments and appreciation of our work. We apologize for the slight delay in our response, as it took us a lot of time to complete the experiments. Below, we address your concerns and suggestions.
>
> **Q1:** *A proper benchmarking of LLMs through uncertainty quantification should include representatives of other methods.*
>
> **A1:** Thanks for this suggestion. In our submission, **we have compared conformal prediction with entropy (or perplexity) in Section 6.6 and Appendix C.7**. The results show that entropy is highly sensitive to the temperature of the softmax function. Moreover, unlike conformal prediction, entropy cannot provide any guarantee on the coverage rate. This is because when measuring uncertainty, entropy doesn't take model accuracy into account. Entropy remains the same when predicted probabilities are permuted, even though prediction accuracy may differ.
>
> To further demonstrate the superiority of conformal prediction, we conduct additional experiments comparing three uncertainty quantification methods: **conformal prediction (CP)**, **entropy**, and **maximal predicted probability ($P_{max}$)**. We adopt the **expected calibration error (ECE)** metric to evaluate the reliability of these methods. The results corresponding to InternLM-7B are presented below:
> | Method  | QA   | RC    | CI    | DRS    | DS   | Avg.    |
> |---------|:--------:|:-------:|:-------:|:-------:|:-------:|:-------:|
> | CP      | **15.83**  | 1.33  | **3.16**  | **12.11** | **9.30**  | **8.35**  |
> | Entropy | **15.83**  | **1.32**  | 3.45  | 12.40 | **9.30**  | 8.46  |
> | $P_{max}$   | **15.83**  | 1.41  | 3.75  | 12.45 | 9.62  | 8.61  |
>
> The results imply that, overall, conformal prediction provides more reliable uncertainty estimation.
>
> **Q2:** *The process of defining a conformal score is itself heuristic.*
>
> **A2:** Thanks for this comment. We will rephrase our statement in the final version to avoid any confusion. The benefit of conformal prediction comes from its ability to take into account both model accuracy and uncertainty. Although there are many possible conformal score functions, the two employed in this study are the most common and widely used ones. Their efficacy has been well demonstrated by existing research. Moreover, these two score functions are highly intuitive and easily interpretable. For example, the LAC function (1 – $P_{true}$) quantifies the difference between the predicted probability ($P_{true}$) and the expected probability (1) of the true option.
>
> **Q3:** *Accuracy and uncertainty go together. The set size metric does not give such intuition. To accurately position the set size metric, I suggest including the AUROC metric.*
>
> **A3:** Thanks for this suggestion. We agree that accuracy and uncertainty should go together. In fact, this is a primary merit of conformal prediction over other uncertainty quantification methods, as it considers model accuracy and uncertainty simultaneously. While our main focus is on the high probability of the prediction set covering the ground-truth, making conformal prediction a robust method for uncertainty quantification, we also demonstrate that **a smaller prediction set (indicating less uncertainty) often correlates with higher accuracy**. We use InternLM-7B as the LLM and conduct experiments on the QA task to illustrate this:
> | Prediction Set Size | LAC   | APS   | Avg.  |
> |---------------------|:-------:|:-------:|:-------:|
> | 1                   | 80.39 | 92.69 | 86.54 |
> | 2                   | 59.77 | 82.21 | 70.99 |
> | 3                   | 40.55 | 63.70 | 52.12 |
> | 4                   | 40.07 | 41.71 | 40.89 |
> | 5                   | 31.50 | 34.92 | 33.21 |
> | 6                   | 13.43 | None  | 13.42 |
>
> In addition, we report the average AUROC and ECE score of different LLMs across five tasks.
> | LLMs          | Prediction Uncertainty (SS) (↓)  | AUROC (%) (↑) | ECE (%) (↓) |
> |---------------|:------:|:-------:|:-------:|
> | Yi-34B        | 1.93 | 69.00 | 8.47  |
> | Qwen-72B      | 2.06 | 68.10 | 8.59  |
> | Qwen-14B      | 2.17 | 64.75 | 7.40  |
> | Llama-2-70B   | 2.16 | 66.17 | 11.17 |
> | DeepSeek-67B  | 2.15 | 68.29 | 14.83 |
> | Yi-6B         | 2.36 | 66.81 | 10.71 |
> | Gemma-7B      | 2.38 | 65.94 | 13.58 |
> | Mistral-7B    | 2.43 | 64.14 | 9.93  |
> | Llama-2-13B   | 2.56 | 62.21 | 11.18 |
> | Qwen-7B       | 2.63 | 66.35 | 9.44  |
> | InternLM-7B   | 3.41 | 60.95 | 8.35  |
> | Llama-2-7B    | 3.09 | 58.26 | 11.98 |
> | DeepSeek-7B   | 3.13 | 61.27 | 8.33  |
> | Qwen-1.8B     | 3.38 | 58.67 | 7.71  |
> | Falcon-40B    | 3.48 | 56.06 | 8.51  |
> | MPT-7B        | 3.57 | 50.69 | 7.15  |
> | Falcon-7B     | 3.75 | 50.39 | 5.99  |
>
> **Q4:** *Does the inclusion of non-ground truth choices (E. none of the above and F. I don’t know) affect uncertainty quantification?*
>
> **A4:** In conformal prediction, we use the prediction set size as a measure of uncertainty, so it's beneficial to have more potential choices. Logically, the presence of more options increases the chance of the prediction set sizes of two LLMs being different, which allows for a more accurate comparison. On the other hand, competent LLMs should be able to identify the correct answer and not be misled by clearly incorrect answers. As shown in **Table 8 in Appendix C.9**, the proportion of test instances where the predicted answer is one of the added two options is quite low, indicating that their inclusion doesn't significantly impact the evaluation.
>
> As shown below, when the LLMs are strong, the average prediction set size can be smaller than 2 even on the challenging summarization task.
> | LLMs        | QA   | RC   | CI   | DRS  | DS   | Avg. |
> |--------------|:------:|:------:|:------:|:------:|:------:|:------:|
> | Yi-34B       | 2.60 | 1.71 | 1.90 | 1.77 | 1.69 | 1.93 |
> | Qwen-72B     | 2.45 | 1.90 | 1.80 | 2.09 | 2.06 | 2.06 |
> | Llama-2-70B   | 2.62 | 1.78 | 1.82 | 2.34 | 2.25 | 2.16 |
> | DeepSeek-67B | 2.65 | 1.54 | 2.43 | 1.89 | 2.25 | 2.15 |

---

> > ### Author Rebuttal · Authors · 2024-08-20
> >
> > **A4 (Cont.):** We further report the average prediction set size when the added two options are removed. It is observed that when LLMs are not strong enough, the average prediction set size can still be larger than 3. Overall, the inclusion of the added options doesn't affect uncertainty quantification significantly.
> > | LLMs         | QA   | RC   | CI   | DRS  | DS   | Avg. |
> > |--------------|:------:|:------:|:------:|:------:|:------:|:------:|
> > | Yi-34B       | 2.06 | 1.42 | 1.52 | 1.49 | 1.58 | 1.61 |
> > | Qwen-72B     | 2.02 | 1.52 | 1.53 | 1.59 | 1.78 | 1.69 |
> > | Llama-2-70B  | 2.30 | 1.51 | 1.69 | 1.88 | 2.07 | 1.89 |
> > | DeepSeek-67B | 2.21 | 1.40 | 2.11 | 1.68 | 2.05 | 1.89 |
> > | Qwen-14B     | 2.39 | 1.41 | 1.54 | 1.67 | 2.04 | 1.81 |
> > | Yi-6B        | 2.41 | 1.54 | 1.77 | 1.99 | 1.76 | 1.89 |
> > | Gemma-7B     | 2.42 | 1.69 | 1.97 | 2.05 | 2.96 | 2.22 |
> > | Mistral-7B   | 2.48 | 1.67 | 2.48 | 2.53 | 2.28 | 2.29 |
> > | Llama-2-13B  | 2.92 | 2.00 | 2.69 | 2.50 | 2.18 | 2.46 |
> > | Qwen-7B      | 2.78 | 1.72 | 2.15 | 2.16 | 2.84 | 2.33 |
> > | InternLM-7B  | 3.34 | 2.08 | 3.47 | 3.13 | 3.69 | 3.14 |
> > | Llama-2-7B   | 3.45 | 2.36 | 3.57 | 3.57 | 2.93 | 3.18 |
> > | DeepSeek-7B  | 3.41 | 2.42 | 3.45 | 3.61 | 3.23 | 3.22 |
> > | Qwen_1.8B    | 3.43 | 2.36 | 3.59 | 3.46 | 3.71 | 3.31 |
> > | Falcon-40B   | 3.46 | 3.20 | 3.70 | 3.72 | 3.66 | 3.55 |
> > | MPT-7B       | 3.73 | 3.68 | 3.75 | 3.75 | 3.78 | 3.74 |
> > | Falcon-7B    | 3.76 | 3.76 | 3.76 | 3.76 | 3.77 | 3.76 |
> >
> > **Q5:** *Conformal prediction restricts the benchmarking to language understanding rather than text generation.*
> >
> > **A5:** Thanks for this comment. Indeed, applying conformal prediction to text generation is a complex task due to the extensive range of potential responses. It's not feasible to compute the probability for each possible response and then use conformal prediction to select a subset. However, many potential responses have a low probability of being generated, which allows us to reduce the selection space by sampling multiple generations.
> >
> > Specifically, **we adopt the TriviaQA dataset [1] (sampling 10K instances) for freeform text generation**. We generate 20 answers for each question. We then utilize the perplexity of each generation as the conformal score function and use exact match to verify the accuracy of the generated answer. The results are displayed in the table below:
> > | LLMs        | Accuracy (Acc, %) (↑) |  Prediction Uncertainty (SS) (↓) | Coverage Rate (CR, %)    |
> > |-------------|:-------:|:------:|:-------:|
> > | Qwen-72B    | 76.45 | 2.63 | 88.92 |
> > | Llama-2-13B | 71.83 | 2.40 | 83.89 |
> > | Qwen-14B    | 66.57 | 3.83 | 82.79 |
> > | Llama-2-7B  | 64.91 | 3.06 | 78.83 |
> > | Qwen-7B     | 59.44 | 5.02 | 77.89 |
> > | DeepSeek-7B | 57.52 | 6.12 | 78.41 |
> > | Falcon-7B   | 55.74 | 6.27 | 76.51 |
> >
> > It can be seen that the prediction set size (SS) varies among models, which could provide some insight into the uncertainty of these models. However, we must note that in this sampling setting, the coverage rate cannot be guaranteed because there might not be a correct answer within the 20 sampled responses. In other words, even if the prediction set size is 20, indicating high model uncertainty, the coverage rate for that instance could still be zero if there are no correct answers present. Nevertheless, it is observed that when the model is stronger, the coverage guarantee requirement is more likely to be satisfied.
> >
> > We aim to develop more effective algorithms for applying conformal prediction to freeform text generation. However, this is a complex task and its challenges are widely acknowledged in the research community.
> >
> > Hope that we have addressed all your concerns. We will also add the potential negative societal impact of our work and describe the existing literature in more detail in the final version. Should you have any further questions, please kindly let us know.
> >
> > [1] https://nlp.cs.washington.edu/triviaqa/

---

> > > ### Author Rebuttal · Authors · 2024-08-28
> > >
> > > Dear Reviewer,
> > >
> > > As the discussion period is approaching the end, may we know if we have answered your questions? Thanks

---

> > > > ### Comment · Reviewer_XKVJ · 2024-08-28
> > > >
> > > > Thank you to the authors for their rebuttal. Most of my concerns are addressed. The inclusion of new results on set size correlating with higher accuracy and conformal results (understandably without a coverage guarantee) beyond multiple choice tasks enriches the work. I increase my score to accept.

---

> > > > > ### Author Rebuttal · Authors · 2024-08-29
> > > > >
> > > > > Much appreciated!!!

---

### Official Review · Reviewer_exNK · 2024-07-25
**A comprehensive pathway to understand uncertainty in LLMs**

**Rating:** 7
**Confidence:** 4
**Clarity:** Yes

**Review:**

- It uncovers that higher accuracy in LLMs does not necessarily correspond with higher certainty, and larger models may exhibit greater uncertainty than smaller ones. Additionally, instruction-finetuning often increases model uncertainty, which has implications for tasks requiring high confidence.
- The study also critically evaluates conformal prediction for uncertainty quantification, noting its limitations such as the need for model output logits and the variability introduced by different conformal score functions.
- Despite these challenges, the paper suggests that conformal prediction remains the most viable method currently available and highlights the need for future research, particularly in the context of multi-modal models.

**Strengths:**

The paper gives a comprehensive understanding of uncertainty in LLMs by revealing nuanced insights into the relationship between model accuracy, scale, and fine-tuning, while critically evaluating the application of conformal prediction.

**Additional Feedback:**

NA

**Correctness:**

The authors have thoroughly outlined all the essential components and methodologies used in this paper, ensuring both transparency and reproducibility.

**Documentation:**

Yes

**Limitations:**

- The requirement for access to model output logits limits the applicability of conformal prediction, making it unsuitable for benchmarking LLMs like OpenAI or Anthropic models, that are accessible only via APIs.
- The uncertainty estimates provided by conformal prediction can vary significantly depending on the conformal score function used, potentially leading to inconsistent assessments of model uncertainty.

**Opportunities For Improvement:**

It would have been great if we could find a way to evaluate the uncertainty of black box LLMs as well

**Relation To Prior Work:**

The paper thoroughly reviews the relevant literature in the field, covering all the works to the best of my knowledge.

**Summary And Contributions:**

The paper presents a comprehensive understanding of uncertainty in Large Language Models (LLMs).

---

> ### Author Rebuttal · Authors · 2024-08-20
>
> Dear Reviewer,
>
> Thank you for your insightful comments and appreciation of our work. We apologize for the slight delay in our response, as it took us a considerable amount of time to complete the experiments. Below, we address your concerns and suggestions.
>
> **Q1:** *Conformal prediction requires access to output logits, making it unsuitable for benchmarking black-box LLMs that are only accessible via APIs.*
>
> **A1:** Thanks for this comment. While getting the exact output logits of black-box LLMs is challenging, we can sample multiple answers and then estimate the probability of each choice. To this end, we perform an experiment on the **MMLU** (the QA task) dataset with GPT-3.5 and GPT-4 as the black-box LLMs. To save cost, we only consider the *Base* prompting strategy. Specifically, we first sample 50 answers for each question and calculate the frequency of each option. Then, we apply label smoothing to prevent zero probabilities. To demonstrate the quality of this approximation, we also report the results of the open-source model Qwen-72B when getting its predictions via sampling and via logits respectively. The results are shown in the following table.
>
> | LLMs                | Coverage Rate (CR, %) | Accuracy (Acc, %) (↑) | Prediction Uncertainty (SS) (↓) |
> |---------------------|:---------------------:|:-----------------:|:---------------------------:|
> | GPT-4               |         90.41         |       81.75       |             1.65            |
> | GPT-3.5             |         89.98         |       62.99       |             3.05            |
> | Qwen-72B (sampling) |         90.54         |       70.29       |             2.43            |
> | Qwen-72B (logits)   |         93.34         |       73.55       |             2.33            |
>
> The average prediction set size (SS) of Qwen-72B (sampling) is relatively close to that of Qwen-72B (logits).
> For each question, we further calculate the **Jensen-Shannon divergence (JSD)** between the predictions of Qwen-72B (sampling) and Qwen-72B (logits). **The average JSD is 0.05, indicating that the two predictions (estimated probability distributions) are highly similar. Therefore, the approximation is of high quality.**
>
>
> **Q2:** *The uncertainty estimates can be influenced by the conformal score function.*
>
> **A2:** This is a valid concern. In our experiments, we have adopted the two most popular score functions and report their average to obtain robust uncertainty quantification.
>
>
> Hope that we have addressed your concerns. Should you have any further questions, kindly let us know.

---

> > ### Comment · Reviewer_exNK · 2024-08-26
> >
> > Thanks for clarifying things.

---

> > > ### Author Rebuttal · Authors · 2024-08-28
> > >
> > > Dear Reviewer,
> > >
> > > Thanks for your reply. We are glad to hear that we have managed to clarify things.

---

### Author Rebuttal · Authors · 2024-08-23

**Summary of our rebuttal**

We sincerely thank all reviewers for their time in reviewing our work and providing many insightful comments, which we found very useful. We have taken these comments into account thoroughly and provided detailed responses with loads of newly added experimental results. These results will undoubtedly further enhance the contributions and quality of our work.

In summary, we have:

- **Added experimental results to better validate the superiority of conformal prediction compared to other uncertainty quantification methods**.

- **Extended our benchmarking to closed-source LLMs with new results regarding GPT-3.5 and GPT-4**.

- **Added experimental results related to freeform question answering by using TriviaQA as the dataset**.

---

### Decision · Program_Chairs · 2024-09-26

**Decision:**

Accept (Poster)

**Comment:**

This paper proposes a benchmark for quantifying the uncertainty of LLMs based on conformal prediction. The experiments include the benchmarking results for nine LLM series on five NLP tasks. This paper shows that a stronger, larger, or instruct-tuned LLM predicts uncertainty better.

The reviewers initially raised concerns such as (1) the requirement for access to model output logits, (2) dependency on the choice of a conformal score function, (3) the reason why conformal prediction is chosen, (4) the limitation of the benchmark (only can measure multiple choice tasks rather than generation tasks)

During the author-reviewer discussion period, the authors showed that they could estimate the probability of different choices even for black-box LLMs. Similarly, they showed that their approach can be applied to freeform text generation tasks (e.g., TriviaQA dataset) using a similar sampling strategy. Finally, additional experimental results with entropy and maximal predicted probability show that in terms of ECE, conformal prediction performs the best among the possible candidates.

After the author-reviewer discussion period, all the reviewers reached a positive consensus. Although a new uncertainty quantification method could potentially change this paper's findings, I think the current form can give good insights to the community. Overall, I recommend acceptance for this paper.